# On the Convergence Theory for Hessian-Free Bilevel Algorithms

**Daouda A. Sow**
Department of ECE
The Ohio State University
sow.53@osu.edu

**Kaiyi Ji**
Department of CSE
University at Buffalo
kaiyiji@buffalo.edu

**Yingbin Liang**
Department of ECE
The Ohio State University
liang.889@osu.edu

## Abstract

Bilevel optimization has arisen as a powerful tool in modern machine learning. However, due to the nested structure of bilevel optimization, even gradient-based methods require second-order derivative approximations via Jacobian- or/and Hessian-vector computations, which can be costly and unscalable in practice. Recently, Hessian-free bilevel schemes have been proposed to resolve this issue, where the general idea is to use zeroth- or first-order methods to approximate the full hypergradient of the bilevel problem. However, we empirically observe that such approximation can lead to large variance and unstable training, but estimating only the response Jacobian matrix as a partial component of the hypergradient turns out to be extremely effective. To this end, we propose a new Hessian-free method, which adopts the zeroth-order-like method to approximate the response Jacobian matrix via taking difference between two optimization paths. Theoretically, we provide the convergence rate analysis for the proposed algorithms, where our key challenge is to characterize the approximation and smoothness properties of the trajectory-dependent estimator, which can be of independent interest. This is the first known convergence rate result for this type of Hessian-free bilevel algorithms. Experimentally, we demonstrate that the proposed algorithms outperform baseline bilevel optimizers on various bilevel problems. Particularly, in our experiment on few-shot meta-learning with ResNet-12 network over the miniImageNet dataset, we show that our algorithm outperforms baseline meta-learning algorithms, while other baseline bilevel optimizers do not solve such meta-learning problems within a comparable time frame.

## 1 Introduction

Bilevel optimization has recently arisen as a powerful tool to capture various modern machine learning problems, including meta-learning [4, 12, 49, 24, 36], hyperparamater optimization [12, 51], neural architecture search [35, 58], signal processing [10], etc. Bilevel optimization generally takes the following mathematical form:

$$\min_{x \in \mathbb{R}^p} \Phi(x) := f(x, y^*(x)), \quad \text{s.t.} \quad y^*(x) = \arg\min_{y \in \mathbb{R}^d} g(x, y), \tag{1}$$

where the outer objective $f : \mathbb{R}^p \times \mathbb{R}^d \to \mathbb{R}$ is continuously differentiable function and the inner objective $g : \mathbb{R}^p \times \mathbb{R}^d \to \mathbb{R}$ is twice differentiable.

Gradient-based methods have served as a popular tool for solving bilevel optimization problems. Two types of approaches have been widely used: the iterative differentiation (ITD) method [7, 11, 51] and the approximate iterative differentiation (AID) method [7, 47, 42]. Due to the bilevel structure of the problem, even such gradient-based methods typically involve *second-order* matrix computations,

36th Conference on Neural Information Processing Systems (NeurIPS 2022).

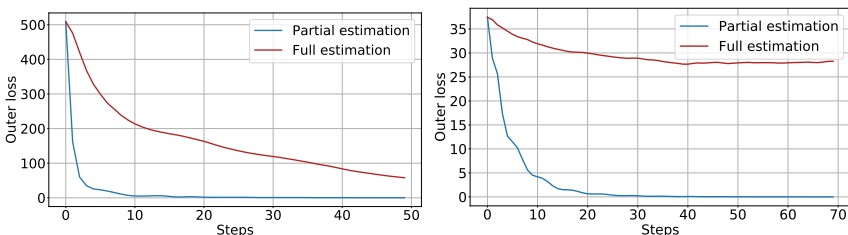

Figure 1: Hyper-representation (HR) with linear (left)/2-layer net (right) embedding model.

because the gradient of $\Phi(x)$ in eq. (1) (which is called *hypergradient*) involves the optimal solution of the inner function. Although the ITD and AID methods often adopt Jacobian- or/and Hessian-vector implementations, the computation can still be very costly in practice for high-dimensional problems with neural networks.

To overcome the computational challenge of current gradient-based methods, a variety of Hessian-free bilevel algorithms have been proposed. For example, popular approaches such as FOMAML [9, 35] ignores the calculations of any second-order derivatives. However, such a trick has no guaranteed performance and may suffer from inferior test performance [1, 8]. In addition, when the outer-level function $f(x, y)$ depends only on $y$ variable, e.g., in some hyperparameter optimization applications [12], it can be shown from eq. (2) that the hypergradient $\nabla\Phi(x)$ vanishes if we eliminate all second-order directives. More recently, several zeroth-order methods [18] have been proposed to approximate the full hypergradient $\nabla\Phi(x)$. In particular, ES-MAML [54] and HOZOG [18] use zeroth-order methods to approximate the **full** hypergraident based on the objective function evaluations.

However, as demonstrated in Figure 1 (also see our further experiments in Section 4), we empirically observe that such full hypergradient estimation can encounter a large variance and inferior performance, whereas a new zeroth-order-like method PZOBO with **partial** hypergradient estimation (as we propose below) performs significantly better.

Encouraged by such an interesting observation, we propose a simple but effective Hessian-free method named PZOBO, which stands for partial zeroth-order-like bilevel optimizer. PZOBO uses a zeroth-order-like approach to approximate only the response Jacobian $\frac{\partial y^*(x)}{\partial x}$ (which is the major computational bottleneck of hypergradient) based on the difference between two gradient-based optimization paths. The remaining terms in hypergradient can simply be calculated by their analytical forms. In this way, PZOBO avoids the large variance of the zeroth-order method for estimating the entire hypergradient, but still enjoys its Hessian-free advantage. We further show that PZOBO admits an easy extension to the large-scale stochastic setting by simply taking small-batch gradients without introducing any complex sub-procedure such as the Neumann Series (NS) type of construction in [23]. Experimentally, PZOBO and its stochastic version PZOBO-S achieve superior performance compared to the current baseline bilevel optimizers, and such an improvement is robust across various bilevel problems. In particular, on the few-shot meta-learning problem over a ResNet-12 network, PZOBO outperforms the state-of-the-art optimization-based meta learners (not necessarily bilevel optimizers), while other bilevel optimizers do not scale to solve such meta-learning problems within a comparable time frame.

Furthermore, our zeroth-order-like response Jacobian estimator in PZOBO takes the difference between two optimization-based trajectories, whereas the vanilla zeroth-order method uses the function value difference at two close points for approximation. Such difference complicates the analysis of PZOBO from three perspectives. (a) Conventional zeroth-order analysis often requires some Lipshitzness properties (e.g., smoothness) of the objective function, whereas they may not hold for the optimization-based output of our response Jacobian estimator. (b) It is unclear if the trajectory-based Hypergradient estimator has bounded error at each iteration, which is critical in the convergence analysis for bilevel optimization [14, 23, 22]. (c) Such characterizations are more challenging for stochastic settings because the randomness along the trajectory needs to be considered.

Theoretically, we provide the convergence rate guarantee for both PZOBO and PZOBO-S. To the best of our knowledge, this is the first known non-asymptotic performance guarantee for Hessian-free bilevel algorithms via zeroth-order-like approximation. Technically, in contrast to the conventional analysis of the zeroth-order estimation on the smoothed blackbox function values, we develop

tools to characterize the bias, variance, smoothness and boundedness of the proposed response Jacobian estimator via a recursive analysis over the gradient-based optimization path, which can be of independent interest for zeroth-order estimation based bilevel optimization.

## 1.1 Related Work

**Bilevel optimization.** Bilevel optimization has been studied for decades since [5]. A variety of bilevel optimization algorithms were then developed, including constraint-based approaches [21, 52, 45], approximate implicit differentiation (AID) approaches [7, 47, 16, 34, 42] and iterative differentiation (ITD) approaches [7, 44, 11, 9, 51, 49, 39]. These methods often suffer from expensive computations of second-order information (e.g., Hessian-vector products). Hessian-free algorithms were recently proposed based on interior-point method [37] and zeroth-order approximation [18, 54]. [38] proposed an initialization auxiliary method to deal with nonconvex inner problem. However, these methods may suffer from either a costly procedure in regularization parameter tuning [37, 38] or a large estimation variance [18, 54] in supervised learning. This paper proposes a simpler and more efficient Hessian-free approach via exploiting the benign structure of the hypergradient and a zeroth-order-like response Jacobian estimation. Recently, the convergence rate has been established for gradient-based (Hessian involved) bilevel algorithms [17, 28, 49, 25]. This paper provides a new convergence analysis for the proposed zeroth-order-like bilevel approach.

**Stochastic bilevel optimization.** [14, 28, 22] proposed stochastic gradient descent (SGD) type bilevel optimization algorithms by employing Neumann Series for Hessian-inverse-vector approximation. Recent works [29, 30, 6, 19, 20, 57] then leveraged momentum-based variance reduction to further reduce the computational complexity of existing SGD-type bilevel optimizers from $\mathcal{O}(\epsilon^{-2})$ to $\mathcal{O}(\epsilon^{-1.5})$. However, these methods often do not scale well in large models due to the Hessian or Hessian inverse computations. In this paper, we propose a stochastic Hessian-free method, which eliminates the computation of all second-order information.

**Bilevel optimization applications.** Bilevel optimization has been employed in various applications such as few-shot meta-learning [53, 12, 49, 59, 24, 27], hyperparameter optimization [11, 43, 51], neural architecture search [35, 58], etc. For example, [43] models the response function itself as a neural network (where each layer involves an affine transformation of hyperparameters) using the Self-Tuning Networks (STNs). An improved and more stable version of STNs was further proposed in [3], which focused on accurately approximating the response Jacobian rather than the response function itself. This paper demonstrates the superior performance of the proposed Hessian-free bilevel optimizer with guaranteed performance in meta-learning and hyperparameter optimization.

**Zeroth-order methods and applications.** Zeroth-order optimization methods have been studied for a long time. For example, [46] proposed an effective zeroth-order gradient estimator via Gaussian smoothing, which was further extended to the stochastic setting by [13]. Such a technique has exhibited great effectiveness in various applications including meta-reinforcement learning [54], hyperparameter optimization [18], adversarial machine learning [26, 40], minimax optimization [41, 56], etc. This paper proposes a novel Jacobian estimator for accelerating bilevel optimization based on an idea similar to zeroth-order estimation via the difference between two optimization paths.

## 2 Proposed Algorithms

### 2.1 Hypergradients

The key step in popular gradient-based bilevel optimizers is the estimation of the hypergradient (i.e., the gradient of the objective with respect to the outer variable $x$), which takes the following form:

$$\nabla \Phi(x) = \nabla_x f(x, y^*(x)) + \mathcal{J}_*(x)^\top \nabla_y f(x, y^*(x)) \tag{2}$$

where the Jacobian matrix $\mathcal{J}_*(x) = \frac{\partial y^*(x)}{\partial x} \in \mathbb{R}^{d \times p}$. Following [42], it can be seen that $\nabla \Phi(x)$ contains two components: the direct gradient $\nabla_x f(x, y^*(x))$ and the indirect gradient $\mathcal{J}_*(x)^\top \nabla_y f(x, y^*(x))$. The direct component can be efficiently computed using the existing automatic differentiation techniques. The indirect component, however, is computationally much more complex, because $\mathcal{J}_*(x)$ takes the form of $\mathcal{J}_*(x) = - \left[ \nabla_y^2 g(x, y^*(x)) \right]^{-1} \nabla_x \nabla_y g(x, y^*(x))$ (if $\nabla_y^2 g(x, y^*(x))$ is invertible), which contains the Hessian inverse and the second-order mixed derivative. Some approaches mitigate the issue by designing Jacobian-vector and Hessian-vector

---

**Algorithm 1** Partial Zeroth-Order-like Bilevel Optimizer (PZOBO)

---

1: **Input:** lower- and upper-level stepsizes $\alpha, \beta > 0$, initializations $x_0 \in \mathbb{R}^p$ and $y_0 \in \mathbb{R}^d$, inner and outer iterations numbers $K$ and $N$, and number of Gaussian vectors $Q$.
2: **for** $k = 0, 1, 2, ..., K$ **do**
3:     Set $y_k^0 = y_0$, $y_{k,j}^0 = y_0$, $j = 1, ..., Q$
4:     **for** $t = 1, 2, ..., N$ **do**
5:         Update $y_k^t = y_k^{t-1} - \alpha \nabla_y g(x_k, y_k^{t-1})$
6:     **end for**
7:     **for** $j = 1, ..., Q$ **do**
8:         Generate $u_{k,j} = \mathcal{N}(0, I) \in \mathbb{R}^p$
9:         **for** $t = 1, 2, ..., N$ **do**
10:           Update $y_{k,j}^t = y_{k,j}^{t-1} - \alpha \nabla_y g\left(x_k + \mu u_{k,j}, y_{k,j}^{t-1}\right)$
11:         **end for**
12:         Compute $\delta_j = \frac{y_{k,j}^N - y_k^N}{\mu}$
13:     **end for**
14:     Compute $\widehat{\nabla}\Phi(x_k) = \nabla_x f(x_k, y_k^N) + \frac{1}{Q} \sum_{j=1}^{Q} \left\langle \delta_j, \nabla_y f(x_k, y_k^N) \right\rangle u_{k,j}$
15:     Update $x_{k+1} = x_k - \beta \widehat{\nabla}\Phi(x_k)$
16: **end for**

---

products [47, 11, 17] to replace second-order computations. But the computation is still costly for high-dimensional bilevel problems such as those with neural network variables. We next introduce the zeroth-order approach which is at the core for designing efficient Hessian-free bilevel optimizers.

## 2.2 Zeroth-Order Approximation

Zeroth-order approximation is a powerful technique to estimate the gradient of a function based on function values, when it is not feasible (such as in black-box problems) or computationally costly to evaluate the gradient. The idea of the zeroth-order method in [46] is to approximate the gradient of a general black-box function $h : \mathbb{R}^n \to \mathbb{R}$ using the following oracle based only on the function values

$$\widehat{\nabla}h(x; u) = \frac{h(x + \mu u) - h(x)}{\mu} u \tag{3}$$

where $u \in \mathbb{R}^n$ is a Gaussian random vector and $\mu > 0$ is the smoothing parameter. Such an oracle can be shown to be an unbiased estimator of the gradient of the smoothed function $\mathbb{E}_u[h(x + \mu u)]$.

## 2.3 Proposed Bilevel Optimizers

Our key idea is to exploit the analytical structure of the hypergradient in eq. (2), where the derivatives $\nabla_x f(x, y^*(x))$ and $\nabla_y f(x, y^*(x))$ can be computed efficiently and accurately, and hence use the zero-order estimator similar to eq. (3) to estimate only the Jacobian $\mathcal{J}_*(x)$, which is the major term posing computational difficulty. In this way, our estimation of the hypergradient can be much more accurate and reliable. In particular, our Jacobian estimator contains two ingredients: **(i)** for a given $x$, apply an algorithm to solve the inner optimization problem and use the output as an approximation of $y^*(x)$; for example, the output $y^N(x)$ of $N$ gradient descent steps of the inner problem can serve as an estimate for $y^*(x)$. Then $\mathcal{J}_N(x) = \frac{\partial y^N(x)}{\partial x}$ serves as an estimate of $\mathcal{J}_*(x)$; and **(ii)** construct the zeroth-order-like Jacobian estimator $\hat{\mathcal{J}}_N(x; u) \in R^{d \times p}$ for $\mathcal{J}_N(x)$ as

$$\hat{\mathcal{J}}_N(x; u) = \frac{y^N(x + \mu u) - y^N(x)}{\mu} u^\top \tag{4}$$

where $u \in \mathbb{R}^p$ is a Gaussian vector with independent and identically distributed (i.i.d.) entries. Then for any vector $v \in \mathbb{R}^d$, the Jacobian-vector product can be efficiently computed using only vector-vector dot product $\hat{\mathcal{J}}_N(x; u)^\top v = \langle \delta(x; u), v \rangle u$, where $\delta(x; u) = \frac{y^N(x + \mu u) - y^N(x)}{\mu} \in \mathbb{R}^d$.

**PZOBO: partial zeroth-order based bilevel optimizer.** For the bilevel problem in eq. (1), we design an optimizer (see Algorithm 1) using the Jacobian estimator in eq. (4), which we call as the PZOBO algorithm. Clearly, the zeroth-order estimator is used only for estimating partial hypergradient. At each step $k$ of the algorithm, PZOBO runs an $N$-step full GD to approximate $y_k^N(x_k)$. PZOBO then

samples $Q$ Gaussian vectors $\{u_{k,j} \in \mathcal{N}(0, I), j = 1, ..., Q\}$, and for each sample $u_{k,j}$, runs an $N$-step full GD to approximate $y_k^N(x_k + \mu u_{k,j})$, and then computes the Jacobian estimator $\hat{\mathcal{J}}_N(x; u_{k,j})$ as in eq. (4). Then the sample average over the $Q$ estimators is used for constructing the following hypergradient estimator for updating the outer variable $x$.

$$\widehat{\nabla}\Phi(x_k) = \nabla_x f(x_k, y_k^N) + \frac{1}{Q}\sum_{j=1}^{Q} \hat{\mathcal{J}}_N^T(x_k; u_{k,j})\nabla_y f(x_k, y_k^N)$$
$$= \nabla_x f(x_k, y_k^N) + \frac{1}{Q}\sum_{j=1}^{Q} \left\langle \delta(x_k; u_{k,j}), \nabla_y f(x_k, y_k^N) \right\rangle u_{k,j}. \tag{5}$$

In our experiments (see Section 4), we choose a small constant-level $Q = \mathcal{O}(1)$ (e.g., $Q = 1$ in most applications with neural nets) due to a much better performance. Our convergence guarantee holds for this case, as shown later.

Computationally, in contrast to the existing AID and ITD bilevel optimizers [47, 12, 17] that contains the complex Hessian- and/or Jacobian-vector product computations, PZOBO has only gradient computations and becomes Hessian-free, and hence is much more efficient as shown in our experiments.

**PZOBO-S: stochastic PZOBO.** In machine learning applications, the loss functions $f, g$ in eq. (1) often take finite-sum forms over given data $\mathcal{D}_{n,m} = \{\xi_i, \zeta_j, i = 1, ..., n, j = 1, ..., m\}$ as below.

$$f(x, y) = \frac{1}{n}\sum_{i=1}^{n} F(x, y; \xi_i), \quad g(x, y) = \frac{1}{m}\sum_{i=1}^{m} G(x, y; \zeta_i) \tag{6}$$

where the sample sizes $n$ and $m$ are typically very large. For such a large-scale scenario, we design a **stochastic** PZOBO bilevel optimizer (see Algorithm 2 in Appendix A), which we call as PZOBO-S.

Differently from Algorithm 1, which applies GD updates to find $y^N(x_k)$, PZOBO-S uses $N$ stochastic gradient descent (SGD) steps to find $\{Y_k^N, Y_{k,1}^N, ..., Y_{k,Q}^N\}$ to the inner problem, each with the outer variable set to be $x_k + \mu u_{k,j}$. Note that all SGD runs follow the same batch sampling path $\{\mathcal{S}_0, ..., \mathcal{S}_{N-1}\}$. The Jacobian estimator $\hat{\mathcal{J}}_N(x_k; u_{k,j})$ can then be computed as in eq. (4). At the outer level, PZOBO-S samples a new batch $\mathcal{D}_F$ independently from the inner batches $\{\mathcal{S}_0, ..., \mathcal{S}_{N-1}\}$ to evaluate the stochastic gradients $\nabla_x F(x_k, Y_k^N; \mathcal{D}_F)$ and $\nabla_y F(x_k, Y_k^N; \mathcal{D}_F)$. The hypergradient $\widehat{\nabla}\Phi(x_k)$ is then estimated as

$$\widehat{\nabla}\Phi(x_k) = \nabla_x F(x_k, Y_k^N; \mathcal{D}_F) + \frac{1}{Q}\sum_{j=1}^{Q} \left\langle \delta(x_k; u_{k,j}), \nabla_y F(x_k, Y_k^N; \mathcal{D}_F) \right\rangle u_{k,j}. \tag{7}$$

## 3 Main Theoretical Results

### 3.1 Technical Assumptions

In this paper, we consider the following types of objective functions.

**Assumption 1.** *The inner function $g(x, y)$ is $\mu_g$-strongly convex with respect to $y$ and the outer function $f(x, y)$ is possibly nonconvex w.r.t. $x$. For the finite-sum case, the same assumption holds for functions $G(x, y; \zeta)$ and $F(x, y; \zeta)$*

The above assumption on $f, g$ has also been adopted in [14, 28, 57]. In fact, many bilevel machine learning problems satisfy this assumption. For example, in few-shot meta-learning, the task-specific parameters are likely the weights of the last classification layer so that the resulting bilevel problem has a strongly-convex inner problem [48, 37].

**Assumption 2.** *Let $w = (x, y)$. The gradient $\nabla g(w)$ is $L_g$-Lipschitz continuous, i.e., for any $w_1, w_2$, $\|\nabla g(w_1) - \nabla g(w_2)\| \leq L_g\|w_1 - w_2\|$; further, the derivatives $\nabla_y^2 g(w)$ and $\nabla_x \nabla_y g(w)$ are $\rho$- and $\tau$-Lipschitz continuous, i.e, $\|\nabla_y^2 g(w_1) - \nabla_y^2 g(w_2)\|_F \leq \rho\|w_1 - w_2\|$ and $\|\nabla_x \nabla_y g(w_1) - \nabla_x \nabla_y g(w_2)\|_F \leq \tau\|w_1 - w_2\|$. The same assumptions hold for $G(w; \zeta)$ in the finite-sum case.*

**Assumption 3.** *Let $w = (x, y)$. The objective $f(w)$ and its gradient $\nabla f(w)$ are $M$- and $L_f$-Lipschitz continuous, i.e., for any $w_1, w_2$, $|f(w_1) - f(w_2)| \leq M\|w_1 - w_2\|$, $\|\nabla f(w_1) - \nabla f(w_2)\| \leq L_f\|w_1 - w_2\|$, which hold for $F(w; \xi)$ in the finite-sum case.*

**Assumption 4.** *For the finite-sum case, the gradient $\nabla G(w; \zeta)$ has bounded variance condition, i.e., $\mathbb{E}_\zeta\|\nabla G(w; \zeta) - \nabla g(w)\|^2 \leq \sigma^2$ for some constant $\sigma \geq 0$.*

## 3.2 Convergence Analysis for PZOBO

Differently from the standard zeroth-order analysis for a blackbox function, here we develop new techniques to analyze the zeroth-order-like Jacobian estimator that depends on the entire inner optimization trajectory, which is unique in bilevel optimization. We first establish the following important proposition which characterizes the Lipshitzness property of the approximate Jacobian matrix $\mathcal{J}_N(x) = \frac{\partial y^N(x)}{\partial x}$.

**Proposition 1.** *Suppose that Assumptions 1 and 2 hold. Let $L_{\mathcal{J}} = \left(1 + \frac{L}{\mu_g}\right)\left(\frac{\tau}{\mu_g} + \frac{\rho L}{\mu_g^2}\right)$, with $L = \max\{L_f, L_g\}$. Then, the Jacobian $\mathcal{J}_N(x)$ is Lipschitz continuous with constant $L_{\mathcal{J}}$:*

$$\left\|\mathcal{J}_N(x_1) - \mathcal{J}_N(x_2)\right\|_F \leq L_{\mathcal{J}}\left\|x_1 - x_2\right\| \quad \forall x_1, x_2 \in \mathbb{R}^p.$$

We next show that the variance of hypergradient estimation can be bounded. The characterization of the estimation bias can be found in Lemma 8 in the appendix.

**Proposition 2.** *Suppose that Assumptions 1, 2, and 3 hold. The hypergradient estimation variance can be upper-bounded as*

$$\mathbb{E}\left\|\widehat{\nabla}\Phi(x_k) - \nabla\Phi(x_k)\right\|^2 \leq \mathcal{O}\left((1 - \alpha\mu_g)^N + \frac{p}{Q} + \mu^2 dp^3 + \frac{\mu^2 dp^4}{Q}\right)$$

*where $\mathbb{E}[\cdot]$ is conditioned on $x_k$ and taken over the Gaussian vectors $\{u_{k,j} : j = 1, ..., Q\}$.*

Proposition 2 upper bounds the hypergradient estimation variance, which mainly arises due to the estimation variance of the Jacobian matrix $\mathcal{J}_*$ by the proposed estimator $\mathcal{J}_N$, via three types of variances: (a) the approximation variance between $\mathcal{J}_N$ and $\mathcal{J}_*$ via inner-loop gradient descent, which decreases exponentially w.r.t. the number $N$ of inner iterations due to the strong convexity of the inner objective; (b) the variance between our estimator and the Jacobian $\mathcal{J}_\mu$ of the smoothed output $\mathbb{E}_u y^N(x_k + \mu u)$, which decreases sublinearly w.r.t. the batch size $Q$, and (c) the variance between the Jacobian $\mathcal{J}_N$ and $\mathcal{J}_\mu$, which can be controlled by the smoothness parameter $\mu$.

By using the smoothness property in Proposition 1 and the upper bound in Proposition 2, we provide the following characterization of the convergence rate for PZOBO.

**Theorem 1** (Convergence of PZOBO). *Suppose that Assumptions 1, 2, and 3 hold. Choose the inner- and outer-loop stepsizes respectively as $\alpha \leq \frac{1}{L}$ and $\beta = \mathcal{O}(\frac{1}{\sqrt{K}})$. Further, set $Q = \mathcal{O}(1)$ and $\mu = \mathcal{O}\left(\frac{1}{\sqrt{Kdp^3}}\right)$. Then, the iterates $x_k$ for $k = 0, ..., K - 1$ of PZOBO satisfy*

$$\frac{1 - \frac{1}{\sqrt{K}}}{K}\sum_{k=0}^{K-1}\mathbb{E}\left\|\nabla\Phi(x_k)\right\|^2 \leq \mathcal{O}\left(\frac{p}{\sqrt{K}} + (1 - \alpha\mu_g)^N\right)$$

Theorem 1 shows that the convergence rate of PZOBO is sublinear with respect to the number $K$ of outer iterations due to the nonconvexity of the outer objective, and linear (i.e., exponentially decay) with respect to the number $N$ of inner iterations due to the strong convexity.

Theorem 1 also indicates the following features that ensures the efficiency of the algorithms. (i) The batch size $Q$ of the Jacobian estimator can be chosen as a constant (in particular $Q = 1$ as in our experiments) so that the computation of the ES estimator is efficient. (ii) The number of inner iterations can be chosen to be small due to its exponential convergence, and hence the algorithm can run efficiently. (iii) PZOBO requires only gradient computations to converge, and eliminates Hessian- and Jacobian-vector products required by the existing AID and ITD based bilevel optimizers ([47, 12, 17]. Thus, PZOBO is more efficient particularly for high-dimensional problems.

## 3.3 Convergence Analysis for PZOBO-S

In this section, we apply the stochastic algorithm PZOBO-S to the finite-sum objective in eq. (6) and analyze its convergence rate. The following proposition establishes an upper bound on the estimation error of Jacobian $\mathcal{J}_*$ by $\mathcal{J}_N = \frac{\partial Y^N}{\partial x}$, where $Y^N$ is the output of $N$ inner SGD updates.

**Proposition 3.** *Suppose that Assumptions 1, 2, and 4 hold. Choose the inner-loop stepsize as $\alpha = \frac{2}{L+\mu_g}$, where $L = \max\{L_f, L_g\}$. Then, we have:*

$$\mathbb{E}\left\|\mathcal{J}_N - \mathcal{J}_*\right\|_F^2 \leq C_\gamma^N \frac{L^2}{\mu_g^2} + \frac{\Gamma}{1 - C_\gamma} + \frac{\lambda(L+\mu_g)^2(1 - \alpha\mu_g)C_\gamma^{N-1}}{(L+\mu_g)^2(1 - \alpha\mu_g) - (L - \mu_g)^2},$$

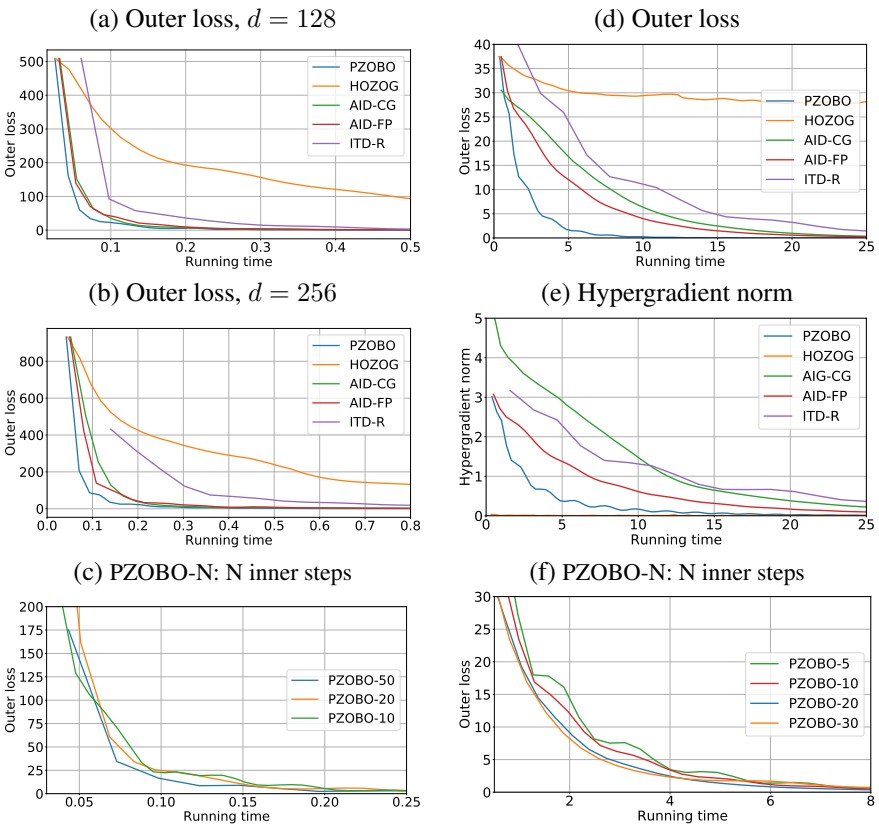

Figure 2: First column: HR with linear embedding model. Second column: HR with two-layer net.

where $\lambda$, $\Gamma$, and $C_\gamma < 1$ are constants (see Appendix J for their precise forms).

We next show that the variance of hypergradient estimation is bounded.

**Proposition 4.** *Suppose that Assumptions 1, 2, 3, and 4 hold. Set the inner-loop stepsize as $\alpha = \frac{2}{L+\mu_g}$ where $L = \max\{L_f, L_g\}$. Then, we have:*

$$\mathbb{E}\big\|\widehat{\nabla}\Phi(x_k) - \nabla\Phi(x_k)\big\|^2 \leq \mathcal{O}\left((1-\alpha\mu_g)^N + \frac{1}{S} + \frac{1}{D_f} + \frac{p}{Q} + \mu^2 dp^3 + \frac{\mu^2 dp^4}{Q}\right)$$

*where $S$ and $D_f$ are the sizes of the inner and outer mini-batches, respectively.*

Based on Propositions 1, 3, and 4, we characterize the convergence rate for PZOBO-S.

**Theorem 2** (Convergence of PZOBO-S). *Suppose that Assumptions 1, 2, 3, and 4 hold. Set the inner- and outer-loop stepsizes respectively as $\alpha = \frac{2}{L+\mu_g}$ and $\beta = \mathcal{O}(\frac{1}{\sqrt{K}})$, where $L = \max\{L_f, L_g\}$. Further, set $Q = \mathcal{O}(1)$, $D_f = \mathcal{O}(1)$, and $\mu = \mathcal{O}\left(\frac{1}{\sqrt{K}dp^3}\right)$. Then, the iterates $x_k, k = 0, ..., K-1$ of PZOBO-S satisfy:*

$$\frac{1-\frac{1}{\sqrt{K}}}{K}\sum_{k=0}^{K-1}\mathbb{E}\big\|\nabla\Phi(x_k)\big\|^2 \leq \mathcal{O}\left(\frac{p}{\sqrt{K}} + (1-\alpha\mu_g)^N + \frac{1}{\sqrt{S}}\right).$$

Comparing to the convergence bound in Theorem 1 for the *deterministic* algorithm PZOBO, Theorem 2 for the *stochastic* algorithm PZOBO-S captures one more sublinearly decreasing error term $\frac{1}{\sqrt{S}}$ due to the sampling of inner batches to estimate the objectives. Note that the sampling of outer batches has been included into the sublinear decay term w.r.t. the number $K$ of outer-loop iterations.

## 4 Experiments

We validate our algorithms in four bilevel problems: **shallow hyper-representation (HR)** with linear/2-layer net embedding model on synthetic data, **deep HR with LeNet network** [32] on MNIST

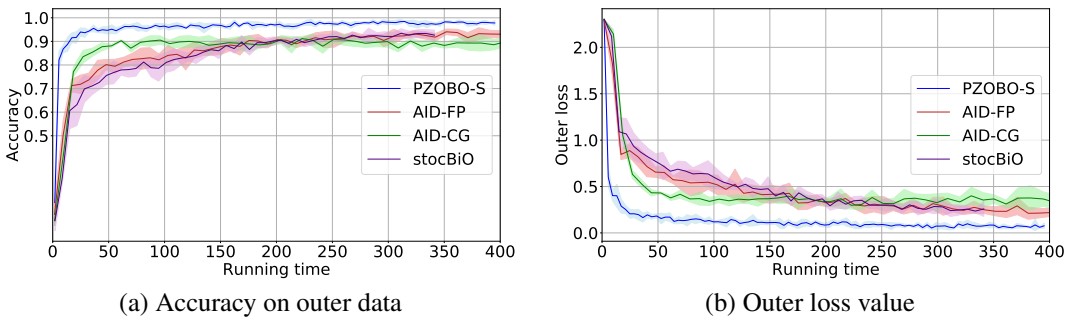

(a) Accuracy on outer data
(b) Outer loss value

Figure 3: Deep HR on the MNIST dataset.

dataset, **few-shot meta-learning** with **ResNet-12** on miniImageNet dataset, and **hyperparameter optimization (HO)** on the 20 Newsgroup dataset. We run all models using a single NVIDIA Tesla P100 GPU. All running time are in seconds.

## 4.1 Shallow Hyper-Representation on Synthetic Data

The hyper-representation (HR) problem [12, 17] searches for a regression (or classification) model following a two-phased optimization process. The inner-level identifies the optimal linear regressor parameters $w$, and the outer level solves for the optimal embedding model (i.e., representation) parameters $\lambda$. Mathematically, the problem can be modeled by the following bilevel optimization:

$$\min_{\lambda \in \mathbb{R}^p} f(\lambda) = \frac{1}{2n_1} \|T(X_1; \lambda)w^* - Y_1\|^2, \text{s.t. } w^* = \underset{w \in \mathbb{R}^d}{\operatorname{argmin}} \frac{1}{2n_2} \|T(X_2; \lambda)w - Y_2\|^2 + \frac{\gamma}{2} \|w\|^2 \quad (8)$$

where $X_2 \in \mathbb{R}^{n_2 \times m}$ and $X_1 \in \mathbb{R}^{n_1 \times m}$ are matrices of synthesized training and validation data, and $Y_2 \in \mathbb{R}^{n_2}$, $Y_1 \in \mathbb{R}^{n_1}$ are the corresponding response vectors. In the case of shallow HR, the embedding function $T(\cdot; \lambda)$ is either a linear transformation or a two-layer network. We generate data matrices $X_1, X_2$ and labels $Y_1, Y_1$ following the same process in [17].

We compare our PZOBO algorithm with the baseline bilevel optimizers AID-FP, AID-CG, ITD-R, and HOZOG (see Appendix E.1 for details about the baseline algorithms and hyperparameters used). Figure 2 show the performance comparison among the algorithms under linear and two-layer net embedding models. It can be observed that for both cases, our proposed method PZOBO converges faster than all the other approaches, and the advantage of PZOBO becomes more significant in Figure 2 (d), which is under a higher-dimensional model of a two-layer net. In particular, PZOBO outperforms the existing ES-based algorithm HOZOG. This is because HOZOG uses the ES technique to approximate the entire hypergradient, which likely incurs a large estimation error. In contrast, our PZOBO exploits the structure of the hypergradient and only estimate the response Jacobian so that the estimation of hypergradient is more accurate. Such an advantage is more evident under a two-layer net model, where HOZOG does not converge as shown in Figure 2 (d). This can be explained by the flat hypergradient norm as shown in Figure 2 (e), which indicates that the hypergradient estimator in HOZOG fails to provide a good descent direction for the outer optimizer. Figure 2 (c) and (f) further show that the convergence of PZOBO does not change substantially with the number $N$ of inner GD steps, and hence tuning of $N$ in practice is not costly.

## 4.2 Deep Hyper-Representation on MNIST Dataset

In order to demonstrate the advantage of our proposed algorithms in neural net models, we perform deep hyper-representation to classify MNIST images by learning an entire LeNet network.The problem formulation is described in Appendix E.3.

Figure 3 compares the classification accuracy on the outer dataset $\mathcal{D}_{\text{out}}$ between our PZOBO-S and other stochastic baseline bilevel optimizers including two AID-based stochastic algorithms AID-FP and AID-CG, and a recently proposed new stochastic bilevel optimizer stocBiO which has been demonstrated to exhibit superior performance. Note that ITD and HOZOG are not included in the comparison, because there have not been stochastic algorithms proposed based on these approaches in the literature yet. Our algorithm PZOBO-S converges with the fastest rate and attains the best

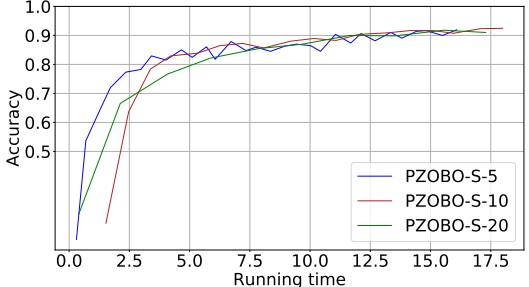

| Algorithm | 67% | 69% |
|---|---|---|
| PZOBO | **2.0** | **2.8** |
| MetaOptNet | 20+ | 20+ |
| ProtNet | 3.5 | 4.4 |
| ANIL | 6.3 | - |
| MAML | 9.7 | - |

Figure 4: **Left plot:** Deep HR on the MNIST dataset with different inner steps number N. PZOBO-S-N means PZOBO-S with N inner steps. **Right table:** Time to reach X% accuracy for few-shot learning experiments with ResNet-12.

accuracy with the lowest variance among all algorithms. Note that PZOBO-S is the only bilevel method that is able to attain the same accuracy of $0.98+$ obtained by the state-of-the-art training of all parameters with one-phased optimization on the MNIST dataset using the same backbone network. All other bilevel methods fail to recover such a level of performance, but instead saturate around an accuracy of $0.93$. Further, the plot in Figure 4 indicates that the convergence of PZOBO-S does not change substantially with the number $N$ of inner SGD steps. This demonstrates the robustness of our method when applied to complex function geometries such as deep nets.

### 4.3 Few-Shot Meta-Learning over MiniImageNet

We study the few-shot image recognition problem, where classification tasks $\mathcal{T}_i, i = 1, ..., m$ are sampled over a distribution $\mathcal{P}_\mathcal{T}$. In particular, we consider the following commonly adopted meta-learning setting (e.g., [48], where all tasks share common embedding features parameterized by $\phi$, and each task $\mathcal{T}_i$ has its task-specific parameter $w_i$ for $i = 1, ..., m$. More specifically, we set $\phi$ to be the parameters of the convolutional part of a deep CNN model (e.g., ResNet-12 network) and $w$ includes the parameters of the last classification layer. All model parameters $(\phi, w)$ are trained following a bilevel procedure. In the inner-loop, the base learner of each task $\mathcal{T}_i$ fixes $\phi$ and minimizes its loss function over a training set $\mathcal{S}_i$ to obtain its adapted parameters $w_i^*$. At the outer stage, the meta-learner computes the test loss for each task $\mathcal{T}_i$ using the parameters $(\phi, w_i^*)$ over a test set $\mathcal{D}_i$, and optimizes the parameters $\phi$ of the common embedding function by minimizing the meta-objective $\mathcal{L}_{\text{meta}}$ over all classification tasks. The detailed problem formulation is given in Appendix E.4.

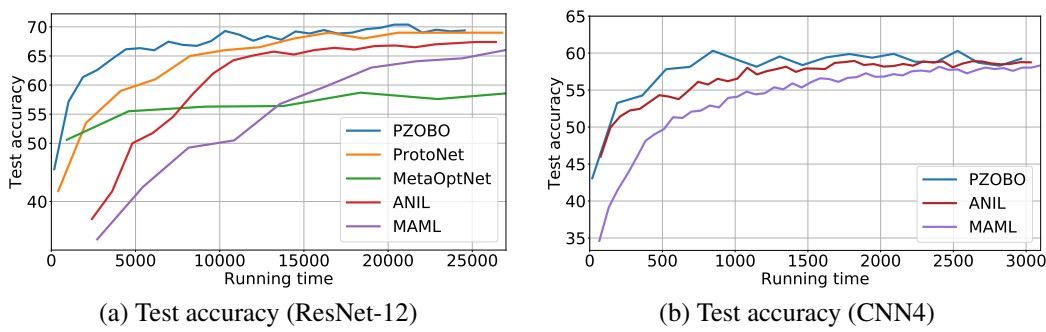

(a) Test accuracy (ResNet-12)  (b) Test accuracy (CNN4)

Figure 5: 5way-5shot few-shot classification on the miniImageNet dataset on single GPU. Left plot: Test accuracy (ResNet-12). Right plot: Test accuracy (CNN4)

We conduct few-shot meta-learning on the miniImageNet dataset [55] using two different backbone networks for feature extraction: ResNet-12 and CNN4 [55]. The dataset and hyperparameter details can be found in Appendix E.4. We compare our algorithm PZOBO with four baseline methods for few-shot meta-learning **MAML** [9], **ANIL** [48], **MetaOptNet** [33], and **ProtoNet** [53]. Other bilevel optimizers are not included into comparison because they do not solve the problem within a comparable time frame. Zeroth-order **ES-MAML** [54] is not included because it exhibits large

variance and cannot reach a desired accuracy. Also note that since ProtoNet and MetaOptNet are usually presented in the ResNet setting, and are not relevant in smaller scale networks, we include them into comparison only for our ResNet-12 experiment. We run their efficient Pytorch Lightning implementations available at the *learn2learn* repository [2].

Figure 5(a) and (b) show that our algorithm PZOBO converges faster than the other baseline meta-learning methods. Also, Comparing Figure 5(a) and (b), the advantage of our method over the baselines **MAML** and **ANIL** becomes more significant as the size of the network increases. Further, the table in Figure 4 shows that MetaOptNet did not reach 69% accuracy after 20 hours of training with ResNet-12 network. In comparison, our PZOBO is able to attain 69% in less than three hours, which is about 1.5 times less than the time taken for ProtoNet to reach the same performance level. Both PZOBO and ProtoNet saturate around 70% accuracy after 10 hours of training.

## 4.4 Hyperparameter Optimization

Hyperparameter optimization (HO) is the problem of finding the set of the best hyperparamters (either representational or regularization parameters) that yield the optimal value of some criterion of model quality (usually a validation loss on unseen data). HO can be posed as a bilevel optimization problem in which the inner problem corresponds to finding the model parameters by minimizing a training loss (usually regularized) for the given hyperparameters and then the outer problem minimizes over the hyperparameters. Due to space limitations, we provide a more complete description of the problem and settings in Appendix F.

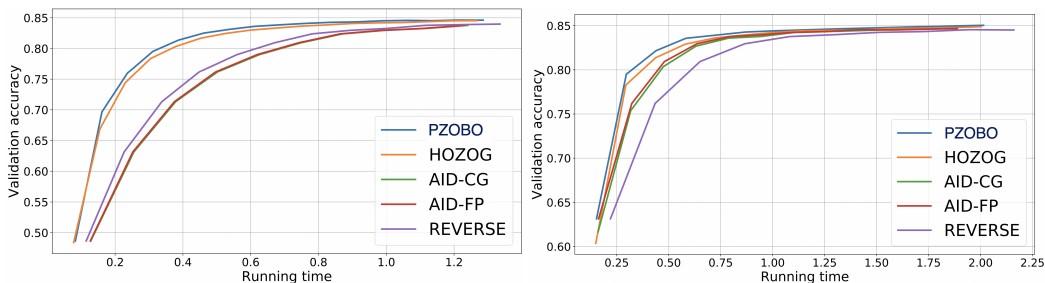

Figure 6: Classification results on 20 Newsgroup dataset. **Left:** number of inner GD step $N = 5$. **Right:** number of inner GD steps $N = 10$.

It can be seen from Figure 6, our method PZOBO slightly outperforms HOZOG and converges faster than the other AID and ITD based approaches. We note that the similar performance for PZOBO and HOZOG can be explained by the fact that in HO, the hypergradient expession in eq. (5) contains only the second term (the first term is zero), which is very close to the approximation in HOZOG method. However, as we have seen in the HR experiments in Figure 2, PZOBO achieves a much better performance than HOZOG. Thus, compared to HOZOG, our PZOBO is much more stable and achieves superior performance across many bilevel problems.

## 5 Conclusion

In this paper, we propose a novel Hessian-free approach for bilevel optimization based on a zeroth-order-like Jacobian estimator. Compared to the existing such types of Hessian-free algorithms, our approach explores the analytical structure of the hypergradient, and hence leads to much more efficient and accurate hypergradient estimation. Thus, our algorithm outperforms existing baselines in various experiments, particularly in high-dimensional problems. We also provide the convergence guarantee and characterize the convergence rate for our proposed algorithms. We anticipate that our approach and analysis will be useful for bilevel optimization and various machine learning applications.

## Acknowledgements

The work of D. Sow and Y. Liang was supported in part by the U.S. National Science Foundation under the grant CCF-1900145 and CNS-2112471.

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
