# Supplementary Material

## A   Stochastic Bilevel Optimizer PZOBO-S

We present the algorithm specification for our proposed **stochastic** bilevel optimizer PZOBO-S.

---

**Algorithm 2** Stochastic PZOBO algorithm (PZOBO-S)

---

1: **Input:** lower- and upper-level stepsizes $\alpha, \beta > 0$, initializations $x_0 \in \mathbb{R}^p$ and $y_0 \in \mathbb{R}^d$, inner and outer iterations numbers $K$ and $N$, and number of Gaussian vectors $Q$.
2: **for** $k = 0, 1, ..., K$ **do**
3:     Set $Y_k^0 = y_0, \quad Y_{k,j}^0 = y_0, j = 1, ..., Q$
4:     Generate $u_{k,j} = \mathcal{N}(0, I) \in \mathbb{R}^p, \quad j = 1, ..., Q$
5:     **for** $t = 1, 2, ..., N$ **do**
6:         Draw a sample batch $\mathcal{S}_{t-1}$
7:         Update $Y_k^t = Y_k^{t-1} - \alpha \nabla_y G(x_k, Y_k^{t-1}; \mathcal{S}_{t-1})$
8:         **for** $j = 1, ..., Q$ **do**
9:             Update $Y_{k,j}^t = Y_{k,j}^{t-1} - \alpha \nabla_y G\left(x_k + \mu u_{k,j}, Y_{k,j}^{t-1}; \mathcal{S}_{t-1}\right)$
10:         **end for**
11:     **end for**
12:     Compute $\delta_j = \frac{Y_{k,j}^N - Y_k^N}{\mu}, \quad j = 1, ..., Q$
13:     Draw a sample batch $\mathcal{D}_F$
14:     Compute $\widehat{\nabla}\Phi(x_k) = \nabla_x F(x_k, Y_k^N; \mathcal{D}_F) + \frac{1}{Q} \sum_{j=1}^Q \left\langle \delta_j, \nabla_y F(x_k, Y_k^N; \mathcal{D}_F) \right\rangle u_{k,j}$
15:     Update $x_{k+1} = x_k - \beta \widehat{\nabla}\Phi(x_k)$
16: **end for**

---

## B   Applicability of Assumptions to Experimental Problems

For the experiments in Sections 4.1 and 4.4, the bilevel problems are relatively simpler with quadratic or logistic loss with a linear classifier. It can be checked that the strong-convexity, smoothness properties are satisfied. For the experiments that involve neural networks, e.g., in deep hyper-representation (Section 4.2) and in meta-learning (Section 4.3), the lower-level problem optimizes only the last-layer parameters and hence is still strongly-convex and smooth. The smoothness of the upper-level problem is not guaranteed, e.g., for ReLU activations, but can still hold true for smoothed ReLU variants.

## C   Comparison to DARTS [35]

### C.1   Discussions

DARTS (Differentiable ARchiTecture Search) [35] was initially proposed for bilevel problems arising from neural architecture search. Our proposed method has substantial differences from the zeroth-order-like estimation in DARTS [35] (i.e., eq. (8) therein). First, DARTS estimates a matrix-vector product, whereas our method estimates the response Jacobian matrix. Second, the estimator in DARTS uses an outer gradient difference evaluated at points with a gap of the inner gradient. In contrast, we use an iterate difference evaluated at two points with gap of a Gaussian random vector. Third, our estimator applies to multiple inner steps with performance guarantee, whereas the estimator in DARTS applies to only the one-step case.

### C.2   Experimental comparison

We compare to DARTS on the bilevel learning problem discussed in [15], in which the lower-level problem is a constraint for guaranteeing good performance on a subset of the upper-level data. Such

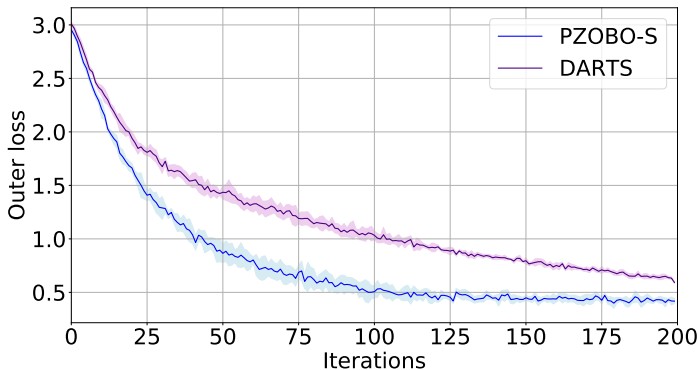

Figure 7: Results on 20 Newsgroup dataset. The batch size is fixed to 128 for both methods. Variances (represented in shadow areas) are computed over 5 different runs.

learning problem can be mathematically formalized as follows

$$\min_{x} \frac{1}{m} \sum_{i=1}^{m} \mathcal{L}\left(x; \xi_i\right) + \frac{1}{n-m} \sum_{i=m+1}^{n} \mathcal{L}\left(y^*(x); \xi_i\right)$$

$$\text{s.t. } y^*(x) = \arg\min_{y} \frac{1}{n-m} \sum_{i=m+1}^{n} \mathcal{L}\left(y; \xi_i\right) + \frac{\lambda}{2} \left\|x - y\right\|^2,$$

where $\mathcal{L}$ is a classification loss function (e.g., cross-entropy) and the regularization term in the lower-level problem encourages the inner variables $y$ to be close to the upper variable $x$.

We conduct the experiments on the 20 News group dataset using linear models for classification and the cross-entropy loss function is used for training. We split the 11314 total training data points into two sets of size 9052 and 2262 (corresponding to a 80% - 20% split), and use the latter set for the inner problem (i.e., $m = 9052$). We remove the news headers in the dataset in order to reduce the dimension of the feature vectors and pre-process the data so as to have feature vectors of dimension $D = 99238$. The dimension of the inner and outer variables $y$ and $x$ correspond to the number of parameters in the weight and bias of the resulting linear model applied to the $D$-dimensional feature vectors, and thus $p = d = 99238 \times 20 + 20$ for 20-ways classification. Due to the relatively large number of datapoints, we apply our stochastic algorithm PZOBO-S and fix the batch size to 128 for both PZOBO-S and DARTS. We plot the outer objective obtained during training of PZOBO-S and DARTS in fig. 7. As it can be seen our method PZOBO-S finds a better descent direction and decreases the objective more importantly at each step.

## D  PZOBO with Different Values of Q

In Figure 8, we compare the performance of our PZOBO algorithm with different choices of $Q$. It can be seen that increasing the number of explorations $Q$ hurts efficiency (because more rounds of inner gradient updates are needed) while providing only marginal performance improvement. It is also shown that the choice of $Q = 1$ achieves the best efficiency with nearly the same accuracy as other choices of $Q > 1$.

## E  Further Specifications for Experiments in Section 4

We note that the smoothing parameter $\mu$ (in Algorithms 1 and 2) was easy to set and a value of $0.1$ or $0.01$ yields a good starting point across all our experiments. The batch size $Q$ (in Algorithms 1 and 2) is fixed to $1$ (i.e., we use one Jacobian oracle) in all our experiments.

### E.1  Specifications on Baseline Bilevel Approaches in Section 4.1

We compare our algorithm PZOBO with the following baseline methods:

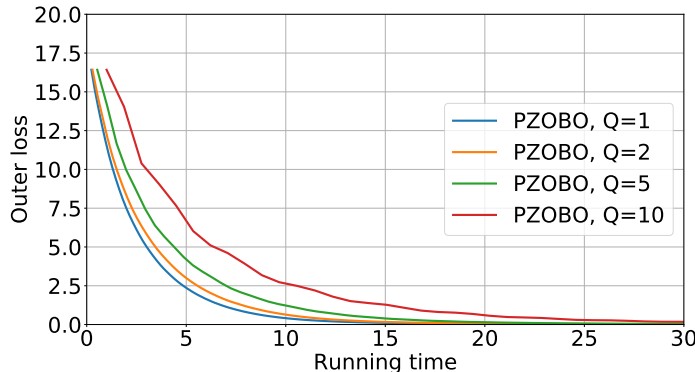

Figure 8: PZOBO with different choices of $Q$ for HR with two-layer net.

- HOZOG [18]: a hyperparameter optimization algorithm that uses evolution strategies to estimate the entire hypergradient (both the direct and indirect component). We use our own implementation for this method.
- AID-CG [17, 49]: approximate implicit differentiation with conjugate gradient. We use its implementation provided at `https://github.com/prolearner/hypertorch`
- AID-FP [17]: approximate implicit differentiation with fixed-point. We experimented with its implementation at the repository `https://github.com/prolearner/hypertorch`
- ITD-R (REVERSE) [11]: an iterative differentiation method that computes hypergradients using reverse mode automatic differention (RMAD). We use its implementation provided at `https://github.com/prolearner/hypertorch`.

### E.2 Hyperparameters Details for Shallow HR Experiments in Section 4.1

For the linear embedding case, we set the smoothing parameter $\mu$ to be $0.01$ for PZOBO and HOZOG. We use the following hyperparameters for all compared methods. The number of inner GD steps is fixed to $N = 20$ with the learning rate of $\alpha = 0.001$. For the outer optimizer, we use Adam [31] with a learning rate of $0.05$. The value of $\gamma$ in eq. (8) is set to be $0.1$. For the two-layer net case, we use $\mu = 0.1$ for PZOBO and HOZOG. For all methods, we set $N = 10$, $\alpha = 0.001$, $\beta = 0.001$, and use Adam with a learning rate of $0.01$ as the outer optimizer.

### E.3 Specifications on Problem Formulation and Baseline Stochastic Algorithms in Section 4.2

The corresponding bilevel problem is given by

$$\min_{\lambda} \mathcal{L}_{\text{out}}(\lambda) := \frac{1}{|\mathcal{D}_{\text{out}}|} \sum_{(x_i, y_i) \in \mathcal{D}_{\text{out}}} \mathcal{L}\left(w^*(\lambda) f(x_i; \lambda), y_i\right)$$

$$\text{s.t.} \quad w^*(\lambda) = \arg\min_{w \in \mathbb{R}^{c \times p}} \mathcal{L}_{\text{in}}(w, \lambda), \quad \mathcal{L}_{\text{in}}(w, \lambda) := \frac{1}{|\mathcal{D}_{\text{in}}|} \sum_{(x_i, y_i) \in \mathcal{D}_{\text{in}}} \mathcal{L}(w f(x_i; \lambda), y_i) + \frac{\beta}{2} \|w\|^2$$

where $f(x_i; \lambda) \in \mathbb{R}^p$ corresponds to features extracted from data point $x_i$, $\mathcal{L}(\cdot, \cdot)$ is the cross-entropy loss function, $c = 10$ is the number of categories, and $\mathcal{D}_{\text{in}}$ and $\mathcal{D}_{\text{out}}$ are data used to compute respectively inner and outer loss functions. Since the sizes of $\mathcal{D}_{\text{out}}$ and $\mathcal{D}_{\text{in}}$ are large in the case of MNIST dataset, we apply the more efficient stochastic algorithm PZOBO-S in Algorithm 2 with a minibatch size $B = 256$ to estimate the inner and outer losses $\mathcal{L}_{\text{in}}$ and $\mathcal{L}_{\text{out}}$.

We compare our method PZOBO-S to the following baseline stochastic bilevel algorithms.

- stocBiO: an approximate implicit differentiation method that uses Neumann Series to estimate the Hessian inverse. We use its implementation available at `https://github.com/JunjieYang97/StocBio`.
- AID-CG-S and AID-FP-S: stochastic versions of AID-CG and AID-FP, respectively. We use their implementations in the repository `https://github.com/prolearner/hypertorch`.

### E.4 Specifications for Few-shot Meta-Learning in Section 4.3

**Problem formulation.** The problem can be expressed as

$$\min_{\phi} \mathcal{L}_{\text{meta}}(\phi, \widetilde{w}^*) := \frac{1}{m} \sum_{i=1}^{m} \mathcal{L}_{\mathcal{D}_i}(\phi, w_i^*)$$

$$\text{s.t.} \ \widetilde{w}^* = \arg\min_{\widetilde{w}} \mathcal{L}_{\text{adapt}}(\phi, \widetilde{w}) := \frac{1}{m} \sum_{i=1}^{m} \mathcal{L}_{\mathcal{S}_i}(\phi, w_i), \tag{9}$$

where we collect all task-specific parameters into $\widetilde{w} = (w_1, ..., w_m)$ and the corresponding minimizers into $\widetilde{w}^* = (w_1^*, ..., w_m^*)$. The functions $\mathcal{L}_{\mathcal{S}_i}(\phi, w_i) = \frac{1}{|\mathcal{S}_i|} \sum_{\zeta \in \mathcal{S}_i} (\mathcal{L}(\phi, w_i; \zeta) + \mathcal{R}(w_i))$ and $\mathcal{L}_{\mathcal{D}_i}(\phi, w_i^*) = \frac{1}{|\mathcal{D}_i|} \sum_{\xi \in \mathcal{D}_i} \mathcal{L}(\phi, w_i^*; \xi)$ correspond respectively to the training and test loss functions for task $\mathcal{T}_i$, with $\mathcal{R}$ a strongly-convex regularizer and $\mathcal{L}$ a classification loss function. In our setting, since the task-specific parameters correspond to the weights of the last linear layer, the inner-level objective $\mathcal{L}_{\text{adapt}}(\phi, \widetilde{w})$ is strongly convex with respect to $\widetilde{w} = (w_1, ..., w_m)$. We note that the problem studied in Section 4.2 can be seen as single-task instances of the more general multi-task learning problem in eq. (9). However, in contrast to the problem in Section 4.2, the datasets $\mathcal{D}_i$ and $\mathcal{S}_i$ are usually small in few-shot learning and full GD can be applied here. Hence, we use ESJ (Algorithm 1) here. Also since the number $m$ of tasks in few-shot classification datasets is often very large, it is preferable to sample a minibatch of i.i.d. tasks by $\mathcal{P}_{\mathcal{T}}$ at each meta (i.e., outer) iteration and update the meta parameters based on these tasks.

**Experimental setup.** The miniImageNet dataset [55] is a large-scale benchmark for few-shot learning generated from ImageNet [50] Russakovsky. The dataset consists of 100 classes with each class containing 600 images of size $84 \times 84$. Following [2], we split the classes into 64 classes for meta-training, 16 classes for meta-validation, and 20 classes for meta-testing. More specfically, we use 20000 tasks for meta-training, 600 tasks for meta-validation, and 600 tasks for meta-testing. We normalize all image pixels by their means and standard deviations over RGB channels and do not perform any additional data augmentation. At each meta-iteration, we sample a batch of 16 training tasks and update the parameters based on these tasks. We set the smoothness parameter to be $\mu = 0.1$ and use $N = 30$ inner steps. We use SGD with a learning rate of $\alpha = 0.01$ as inner optimizer and Adam with a learning rate of $\beta = 0.01$ as outer (meta) optimizer.

## F Experiments on Hyperparameter Optimization

Hyperparameter optimization (HO) is the problem of finding the set of the best hyperparamters (either representational or regularization parameters) that yield the optimal value of some criterion of model quality (usually a validation loss on unseen data). HO can be posed as a bilevel optimization problem in which the inner problem corresponds to finding the model parameters by minimizing a training loss (usually regularized) for the given hyperparameters and then the outer problem minimizes over the hyperparameters. Hence, HO can be mathematically expressed as follows

$$\min_{\lambda} \mathcal{L}_{\text{val}}(\lambda) := \frac{1}{|\mathcal{D}_{\text{val}}|} \sum_{\xi \in \mathcal{D}_{\text{val}}} \mathcal{L}(w^*(\lambda); \xi)$$
$$\text{s.t.} \quad w^*(\lambda) = \arg\min_{w} \mathcal{L}_{\text{tr}}(w, \lambda) := \frac{1}{|\mathcal{D}_{\text{tr}}|} \sum_{\zeta \in \mathcal{D}_{\text{tr}}} (\mathcal{L}(w, \lambda; \zeta) + \mathcal{R}(w, \lambda)), \tag{10}$$

where $\mathcal{L}$ is a loss function (e.g., logistic loss), $\mathcal{R}(w, \lambda)$ is a regularizer, and $\mathcal{D}_{tr}$ and $\mathcal{D}_{val}$ are respectively training and validation data. Note that the loss function used to identify hyperparameters must be different from the one used to find model parameters; otherwise models with higher complexities would be always favored. This is usually achieved in HO by using different data splits (here $\mathcal{D}_{val}$ and $\mathcal{D}_{tr}$) to compute validation and training losses, and by adding a regularizer term on the training loss.

Following [11, 17], we perform classification on the 20 Newsgroup dataset, where the classifier is modeled by an affine transformation, the cost function $\mathcal{L}$ is the cross-entrpy loss, and $\mathcal{R}(w, \lambda)$ is a strongly-convex regularizer. We set one $l_2$-regularization hyperparameter for each weight in $w$, so that $\lambda$ and $w$ have the same size.

For PZOBO and HOZOG, we use GD with a learning rate of 100 and a momentum of 0.9 to perform the inner updates. The outer learning rate is set to be 0.02. We set the smoothing parameter ($\mu$ in

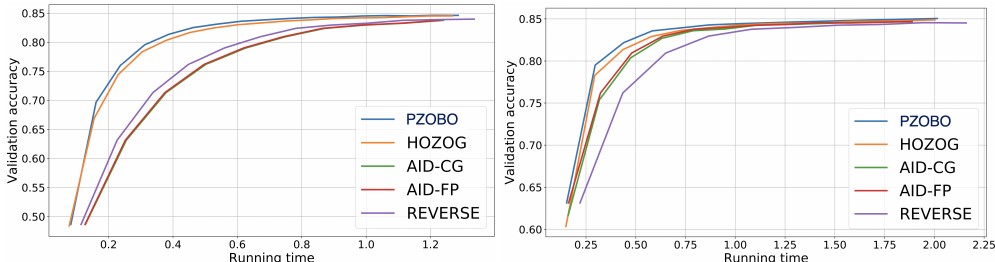

Figure 9: Classification results on 20 Newsgroup dataset. **Left:** number of inner GD step $N = 5$. **Right:** number of inner GD steps $N = 10$.

Algorithm 1) to be $0.01$. For AID-FP, AID-CG, and REVERSE we use the suggested hyperparameters in their implementations accompanying the paper [17].

It can be seen from Figure 9, our method PZOBO slightly outperforms HOZOG and converges faster than the other AID and ITD based approaches. We note that the similar performance for PZOBO and HOZOG can be explained by the fact that in HO, the hypergradient expression in eq. (5) contains only the second term (the first term is zero), which is very close to the approximation in HOZOG method. However, as we have seen in the other experiments (not for HO), PZOBO is a more robust and stable bilevel optimizer than HOZOG, and it achieves good performance across many bilevel problems.

## G    Problem Dimensions of All Experiments

For the hyper-representation problems, the dimension of the hyperparameter vector (outer variable) corresponds to the number of parameters in the embedding model. For example, for the deep hyper-representation problem, the dimension of the hyperparameter vector $p$ corresponds to the number of weights in the backbone LeNet network. For the shallow hyper-representation problem, the value of $p$ depends on the input dimension and representations dimension d. For this experiments we always set the input dimension to be half of the representations dimension d, which corresponds to $p = d \times d/2$ for the linear embedding settings.

For meta-learning experiments, the dimension of the hyperparameter vector (or outer variable) corresponds to the number of weights in the common embedding model (which is either ResNet12 or CNN4). For hyperparameter optimization experiments, we used one regularization parameter for each weight in the linear classification model, which means that we have as many hyperparameters as parameters. The dimension of the input features in the 20 Newsgroup dataset is  130K and we simply use a linear model without bias term, which corresponds to the number of parameters (and thus hyperparameters) $p =  130K \times 20$.

## H    Supporting Technical Lemmas

In this section, we provide auxiliary lemmas that are used for proving the convergence results for the algorithms PZOBO and PZOBO-S.

In the following proofs, we let $L = \max\{L_g, L_f\}$ and $D$ be such that $\|y^*(x)\| \leq D$.

First we recall that for any two matrices $A \in \mathbb{R}^{m \times r}$ and $B \in \mathbb{R}^{r \times n}$, we have the following upper-bound on the Frobenius norm of their product,

$$\|AB\|_F \leq \|A\|\|B\|_F. \tag{11}$$

The following lemma follows directly from the Lipschitz properties in Assumptions 2 and 3.

**Lemma 1.** *Suppose that Assumptions 2 and 3 hold. Then, the stochastic derivatives $\nabla F(x, y; \xi)$, $\nabla_x \nabla_y G(x, y; \xi)$, and $\nabla_y^2 G(x, y; \xi)$ have bounded variances, i.e., for any $(x, y)$ and $\xi$ we have:*

- $\mathbb{E}_\xi \|\nabla F(x, y; \xi) - \nabla f(x, y)\|^2 \leq M^2;$

- $\mathbb{E}_{\xi} \big\| \nabla_x \nabla_y G(x, y; \xi) - \nabla_x \nabla_y g(x, y) \big\|_F^2 \leq L^2;$
- $\mathbb{E}_{\xi} \big\| \nabla_y^2 G(x, y; \xi) - \nabla_y^2 g(x, y) \big\|_F^2 \leq L^2.$

Using Lemma 2.2 in [14], the following lemma characterizes the Lipschitz property of the gradient of the total objective $\Phi(x) = f(x, y^*(x))$.

**Lemma 2.** *Suppose that Assumptions 1, 2, and 3 hold. Then, we have:*

$$\big\| \nabla \Phi(x_2) - \nabla \Phi(x_1) \big\| \leq L_{\Phi} \big\| x_2 - x_1 \big\| \qquad \forall x_1 \in \mathbb{R}^p, x_2 \in \mathbb{R}^p,$$

*where the constant $L_{\Phi} = L + \frac{2L^2 + \tau M^2}{\mu_g} + \frac{\rho L M + L^3 + \tau M L}{\mu_g^2} + \frac{\rho L^2 M}{\mu_g^3}$.*

We next provide some essential properties of the zeroth-order gradient oracle in eq. (3), due to [46].

**Lemma 3.** *Let $h : \mathbb{R}^n \to \mathbb{R}$ be a differentiable function with L-Lipsctitz gradient. Define its Gaussian smooth approximation $h_{\mu}(x) = \mathbb{E}_u [h(x + \mu u)]$, where $\mu > 0$ and $u \in \mathbb{R}^n$ is a standard Gaussian random vector. Then, $h_{\mu}$ is differentiable and we have:*

- *The gradient of $h_{\mu}$ takes the form*

$$\nabla h_{\mu}(x) = \mathbb{E}_u \frac{h(x + \mu u) - h(x)}{\mu} u.$$

- *For any $x \in \mathbb{R}^n$,*

$$\big\| \nabla h_{\mu}(x) - \nabla h(x) \big\| \leq \frac{\mu}{2} L(n + 3)^{3/2}.$$

- *For any $x \in \mathbb{R}^n$,*

$$\mathbb{E}_u \left\| \frac{h(x + \mu u) - h(x)}{\mu} u \right\|^2 \leq 4(n + 4) \big\| \nabla h_{\mu}(x) \big\|^2 + \frac{3}{2} \mu^2 L^2 (n + 5)^3.$$

Note the first item in Lemma 3 implies that the oracle in eq. (3) is in indeed an unbiased estimator of the gradient of the smoothed function $h_{\mu}$.

**Lemma 4.** *Suppose that Assumptions 1 and 2 hold. The Jacobian $\mathcal{J}_* = \frac{\partial y^*(x)}{\partial x}$ has bounded norm:*

$$\big\| \mathcal{J}_* \big\|_F \leq \frac{L}{\mu_g}. \tag{12}$$

*Proof of Lemma 4.* From the first order optimality condition of $y^*(x)$, we have $\nabla_y g(x, y^*(x)) = 0$. Hence, the Implicit Function Theorem implies:

$$\mathcal{J}_* = - \big[ \nabla_y^2 g(x, y^*(x)) \big]^{-1} \nabla_x \nabla_y g(x, y^*(x)). \tag{13}$$

Taking norms and applying eq. (11) together with Assumptions 1 and 2 yield the desired result

$$\big\| \mathcal{J}_* \big\|_F \leq \big\| \nabla_x \nabla_y g(x, y^*(x)) \big\|_F \big\| \big[ \nabla_y^2 g(x, y^*(x)) \big]^{-1} \big\| \leq \frac{L}{\mu_g}. \tag{14}$$

$\square$

**Lemma 5.** *Suppose that Assumptions 1 and 2 hold. The Jacobian $\mathcal{J}_N = \frac{\partial y_k^N}{\partial x_k}$ has bounded norm:*

$$\big\| \mathcal{J}_N \big\|_F \leq \frac{L}{\mu_g}. \tag{15}$$

*Proof of Lemma 5.* The inner loop gradient descent updates writes

$$y_k^t = y_k^{t-1} - \alpha \nabla_y g(x_k, y_k^{t-1}), \quad t = 1, \ldots, N.$$

Taking derivatives w.r.t. $x_k$ yields

$$\mathcal{J}_t = \mathcal{J}_{t-1} - \alpha \nabla_x \nabla_y g(x_k, y_k^{t-1}) - \alpha \mathcal{J}_{t-1} \nabla_y^2 g(x_k, y_k^{t-1})$$

$$= \mathcal{J}_{t-1}\left(I - \alpha\nabla_y^2 g\left(x_k, y_k^{t-1}\right)\right) - \alpha\nabla_x\nabla_y g\left(x_k, y_k^{t-1}\right).$$

Telescoping over $t$ from 1 to $N$ yields

$$\mathcal{J}_N = \mathcal{J}_0 \prod_{t=0}^{N-1}\left(I - \alpha\nabla_y^2 g\left(x_k, y_k^t\right)\right) - \alpha\sum_{t=0}^{N-1}\nabla_x\nabla_y g\left(x_k, y_k^t\right)\prod_{m=t+1}^{N-1}\left(I - \alpha\nabla_y^2 g\left(x_k, y_k^m\right)\right)$$

$$= -\alpha\sum_{t=0}^{N-1}\nabla_x\nabla_y g\left(x_k, y_k^t\right)\prod_{m=t+1}^{N-1}\left(I - \alpha\nabla_y^2 g\left(x_k, y_k^m\right)\right). \tag{16}$$

Hence, we have

$$\left\|\mathcal{J}_N\right\|_F \leq \alpha\sum_{t=0}^{N-1}\left\|\nabla_x\nabla_y g\left(x_k, y_k^t\right)\right\|_F\prod_{m=t+1}^{N-1}\left(I - \alpha\nabla_y^2 g\left(x_k, y_k^m\right)\right)\|$$

$$\overset{(i)}{\leq} \alpha\sum_{t=0}^{N-1}L\prod_{m=t+1}^{N-1}\left\|I - \alpha\nabla_y^2 g\left(x_k, y_k^m\right)\right\|$$

$$\overset{(ii)}{\leq} \alpha L\sum_{t=0}^{N-1}(1 - \alpha\mu_g)^{N-1-t}$$

$$= \alpha L\sum_{t=0}^{N-1}(1 - \alpha\mu_g)^t \leq \frac{L}{\mu_g}$$

where $(i)$ follows from Assumption 2 and $(ii)$ applies the strong-convexity of function $g(x, \cdot)$. This completes the proof. $\qquad\square$

**Lemma 6.** *Suppose that Assumptions 1 and 2 hold. Then, the Jacobian $\mathcal{J}_N = \frac{\partial Y_k^N}{\partial x_k}$ in the stochastic algorithm PZOBO-S has bounded norm, as shown below.*

$$\left\|\mathcal{J}_N\right\|_F \leq \frac{L}{\mu_g}. \tag{17}$$

*Proof of Lemma 6.* The proof follows similarly to Lemma 5. $\qquad\square$

### H.1 Proof of Proposition 1

**Proposition 5** (Restatement of Proposition 1). *Suppose that Assumptions 1 and 2 hold. Define the constant*

$$L_{\mathcal{J}} = \left(1 + \frac{L}{\mu_g}\right)\left(\frac{\tau}{\mu_g} + \frac{\rho L}{\mu_g^2}\right). \tag{18}$$

*Then, the Jacobian $\mathcal{J}_N(x) = \frac{\partial y^N(x)}{\partial x}$ is $L_{\mathcal{J}}$-Lipschitz with respect to $x$ under the Frobenious norm:*

$$\left\|\mathcal{J}_N(x_1) - \mathcal{J}_N(x_2)\right\|_F \leq L_{\mathcal{J}}\left\|x_1 - x_2\right\| \qquad \forall x_1 \in \mathbb{R}^p, x_2 \in \mathbb{R}^p. \tag{19}$$

*Proof of Proposition 1.* Using eq. (16), we have

$$\mathcal{J}_N(x) = -\alpha\sum_{t=0}^{N-1}\nabla_x\nabla_y g\left(x, y^t(x)\right)\prod_{m=t+1}^{N-1}\left(I - \alpha\nabla_y^2 g\left(x, y^m(x)\right)\right), \quad x \in \mathbb{R}^p.$$

Hence, for $x_1 \in \mathbb{R}^p$ and $x_2 \in \mathbb{R}^p$, we have

$$\left\|\mathcal{J}_N(x_1) - \mathcal{J}_N(x_2)\right\|_F$$

$$= \alpha\left\|\sum_{t=0}^{N-1}\nabla_x\nabla_y g\left(x_1, y^t(x_1)\right)\prod_{m=t+1}^{N-1}\left(I - \alpha\nabla_y^2 g\left(x_1, y^m(x_1)\right)\right)\right.$$

$$- \sum_{t=0}^{N-1} \nabla_x \nabla_y g \left( x_2, y^t(x_2) \right) \prod_{m=t+1}^{N-1} \left( I - \alpha \nabla_y^2 g \left( x_2, y^m(x_2) \right) \right) \Big\|_F$$

$$\leq \alpha \sum_{t=0}^{N-1} \Big\| \nabla_x \nabla_y g \left( x_1, y^t(x_1) \right) \prod_{m=t+1}^{N-1} \left( I - \alpha \nabla_y^2 g \left( x_1, y^m(x_1) \right) \right)$$

$$- \nabla_x \nabla_y g \left( x_2, y^t(x_2) \right) \prod_{m=t+1}^{N-1} \left( I - \alpha \nabla_y^2 g \left( x_2, y^m(x_2) \right) \right) \Big\|_F$$

$$\overset{(i)}{\leq} \alpha \sum_{t=0}^{N-1} \left\| \nabla_x \nabla_y g \left( x_1, y^t(x_1) \right) \right\|_F \left\| A_t(x_1) - A_t(x_2) \right\|$$

$$+ \alpha \sum_{t=0}^{N-1} \left\| A_t(x_2) \right\| \left\| \nabla_x \nabla_y g \left( x_1, y^t(x_1) \right) - \nabla_x \nabla_y g \left( x_2, y^t(x_2) \right) \right\|_F \tag{20}$$

where we define $A_t(x) = \prod_{m=t+1}^{N-1} \left( I - \alpha \nabla_y^2 g \left( x, y^m(x) \right) \right)$ and $(i)$ follows from eq. (11).

Next we upper bound the quantity $\left\| A_t(x_1) - A_t(x_2) \right\|$, as shown below.

$$\left\| A_t(x_1) - A_t(x_2) \right\| \leq \Big\| \alpha \left( \nabla_y^2 g \left( x_2, y^{t+1}(x_2) \right) - \nabla_y^2 g \left( x_2, y^{t+1}(x_2) \right) \right) A_{t+1}(x_1)$$

$$+ \left( I - \alpha \nabla_y^2 g \left( x_2, y^{t+1}(x_2) \right) \right) \left( A_{t+1}(x_1) - A_{t+1}(x_2) \right) \Big\|$$

$$\leq \alpha \left\| A_{t+1}(x_1) \right\| \left\| \nabla_y^2 g \left( x_1, y^{t+1}(x_1) \right) - \nabla_y^2 g \left( x_2, y^{t+1}(x_2) \right) \right\|$$

$$+ \left\| I - \alpha \nabla_y^2 g \left( x_2, y^{t+1}(x_2) \right) \right\| \left\| A_{t+1}(x_1) - A_{t+1}(x_2) \right\|$$

$$\leq (1 - \alpha \mu_g) \left\| A_{t+1}(x_1) - A_{t+1}(x_2) \right\|$$

$$+ \alpha \rho (1 + \frac{L}{\mu_g})(1 - \alpha \mu_g)^{N-t-2} \left\| x_1 - x_2 \right\|, \tag{21}$$

where the last inequality follows from Lemma 5 and Assumptions 1 and 2.

Telescoping eq. (21) over $t$ yields

$$\left\| A_t(x_1) - A_t(x_2) \right\| \leq (1 - \alpha \mu_g)^{N-t-2} \left\| A_{N-2}(x_1) - A_{N-2}(x_2) \right\|$$

$$+ \sum_{m=0}^{N-t-3} \alpha \rho (1 + \frac{L}{\mu_g})(1 - \alpha \mu_g)^{N-t-2-m}(1 - \alpha \mu_g)^m \left\| x_1 - x_2 \right\|$$

$$= (1 - \alpha \mu_g)^{N-t-2} \left\| \nabla_y^2 g \left( x_1, y^{N-1}(x_1) \right) - \nabla_y^2 g \left( x_2, y^{N-1}(x_2) \right) \right\|$$

$$+ \sum_{m=0}^{N-t-3} \alpha \rho (1 + \frac{L}{\mu_g})(1 - \alpha \mu_g)^{N-t-2} \left\| x_1 - x_2 \right\|$$

$$\overset{(i)}{\leq} \alpha \rho (1 + \frac{L}{\mu_g})(1 - \alpha \mu_g)^{N-t-2} \left\| x_1 - x_2 \right\|$$

$$+ (N - t - 2) \alpha \rho (1 + \frac{L}{\mu_g})(1 - \alpha \mu_g)^{N-t-2} \left\| x_1 - x_2 \right\|$$

$$= \alpha \rho (1 + \frac{L}{\mu_g})(N - t - 1)(1 - \alpha \mu_g)^{N-t-2} \left\| x_1 - x_2 \right\|, \tag{22}$$

where $(i)$ follows from Lemma 5 and Assumption 2. Replacing eq. (22) in eq. (20) and using Assumption 2, we have

$$\left\| \mathcal{J}_N(x_1) - \mathcal{J}_N(x_2) \right\|_F$$

$$\leq \alpha \sum_{t=0}^{N-1} L \alpha \rho (1 + \frac{L}{\mu_g})(N - t - 1)(1 - \alpha \mu_g)^{N-t-2} \left\| x_1 - x_2 \right\|$$

$$+ \alpha \sum_{t=0}^{N-1} \tau (1 + \frac{L}{\mu_g})(1 - \alpha \mu_g)^{N-t-1} \left\| x_1 - x_2 \right\|$$

$$\leq \alpha^2 L\rho(1 + \frac{L}{\mu_g})\|x_1 - x_2\| \sum_{m=0}^{N-1} m(1 - \alpha\mu_g)^{m-1} + \frac{\tau}{\mu_g}(1 + \frac{L}{\mu_g})\|x_1 - x_2\|$$

$$\leq \frac{\rho L}{\mu_g^2}(1 + \frac{L}{\mu_g})\|x_1 - x_2\| + \frac{\tau}{\mu_g}(1 + \frac{L}{\mu_g})\|x_1 - x_2\| \tag{23}$$

where we use $\sum_{m=0}^{N-1} mx^{m-1} \leq \frac{1}{\alpha^2 \mu_g^2}$ in eq. (23), which can be obtained by taking derivatives for the expression $\sum_{m=0}^{N-1} x^m$ with respect to $x$. Hence, rearranging and using the definition of $L_{\mathcal{J}}$ in eq. (18) finishes the proof. □

**Lemma 7.** *Suppose that Assumptions 1 and 2 hold. Define the constant*

$$L_{\mathcal{J}} = \left(1 + \frac{L}{\mu_g}\right)\left(\frac{\tau}{\mu_g} + \frac{\rho L}{\mu_g^2}\right).$$

*Then, the Jacobian* $\mathcal{J}_N(x) = \frac{\partial Y^N(x;\cdot)}{\partial x}$ *is* $L_{\mathcal{J}}$*-Lipschitz with respect to* $x$ *under the Frobenius norm:*

$$\left\|\mathcal{J}_N(x_1;\cdot) - \mathcal{J}_N(x_2;\cdot)\right\|_F \leq L_{\mathcal{J}}\|x_1 - x_2\| \qquad \forall x_1 \in \mathbb{R}^p, x_2 \in \mathbb{R}^p.$$

*Proof of Lemma 7.* The proof follows similarly to that for Proposition 1. □

# I Proofs for Deterministic Bilevel Optimization

For notation convenience, we define the following quantities:

$$\hat{\mathcal{J}}_{N,j} = \hat{\mathcal{J}}_N(x_k, u_j) = \begin{pmatrix} \frac{y_1^N(x_k + \mu u_j) - y_1^N(x_k)}{\mu} u_j^\top \\ \vdots \\ \frac{y_d^N(x_k + \mu u_j) - y_d^N(x_k)}{\mu} u_j^\top \end{pmatrix}, \quad \mathcal{J}_N = \frac{\partial y_k^N}{\partial x_k}, \quad \mathcal{J}_* = \frac{\partial y_k^*}{\partial x_k}, \tag{24}$$

where $u_j \in \mathbb{R}^p, j = 1, \ldots, Q$ are standard Gaussian vectors. Let $y_{i,\mu}^N(x_k)$ be the Gaussian smooth approximation of $y_i^N(x_k)$. We collect $y_{i,\mu}^N(x_k)$ for $i = 1, \ldots, d$ together as a vector $y_\mu^N(x_k)$, which is the Gaussian approximation of the vector $y^N(x_k)$. If $\mu > 0$, $y_\mu^N(x_k)$ is differentiable and we let $\mathcal{J}_\mu$ be the Jocobian given by

$$\mathcal{J}_\mu = \frac{\partial y_\mu^N(x_k)}{\partial x_k}. \tag{25}$$

We approximate $\frac{\partial y_k^N}{\partial x_k}$ using the average zeroth-order estimator given by $\hat{\mathcal{J}}_N = \frac{1}{Q}\sum_{j=1}^Q \hat{\mathcal{J}}_{N,j}$. The hypergradient is then approximated as

$$\widehat{\nabla}\Phi(x_k) = \nabla_x f(x_k, y_k^N) + \hat{\mathcal{J}}_N^\top \nabla_y f(x_k, y_k^N)$$

$$= \nabla_x f(x_k, y_k^N) + \frac{1}{Q}\sum_{j=1}^Q \hat{\mathcal{J}}_{N,j}^\top \nabla_y f(x_k, y_k^N). \tag{26}$$

Let $\delta_j = \frac{y^N(x_k + \mu u_j) - y^N(x_k)}{\mu}$ and let $\delta_{i,j}$ be the $i$-th component of $\delta_j$. Hence, we have

$$\hat{\mathcal{J}}_{N,j} = \begin{pmatrix} \delta_{1,j} u_j^\top \\ \delta_{2,j} u_j^\top \\ \vdots \\ \delta_{d,j} u_j^\top \end{pmatrix},$$

$$\hat{\mathcal{J}}_{N,j}^\top \nabla_y f(x_k, y_k^N) = \begin{pmatrix} \delta_{1,j} u_j & \delta_{2,j} u_j & \cdots & \delta_{d,j} u_j \end{pmatrix} \nabla_y f(x_k, y_k^N)$$

$$= \langle \delta_j, \nabla_y f(x_k, y_k^N) \rangle u_j. \tag{27}$$

Using eq. (26) and eq. (27), the estimator for the hypergradient can thus be computed as

$$\widehat{\nabla}\Phi(x_k) = \nabla_x f(x_k, y_k^N) + \frac{1}{Q}\sum_{j=1}^Q \langle \delta_j, \nabla_y f(x_k, y_k^N) \rangle u_j.$$

## I.1 Proof of Proposition 2

**Proposition 6** (Formal Statement of Proposition 2). *Suppose that Assumptions 1, 2, and 3 hold. Then, the variance of hypergradient estimation can be upper-bounded as*

$$
\mathbb{E}\big\|\widehat{\nabla}\Phi(x_k) - \nabla\Phi(x_k)\big\|^2 \leq 2L^2 D^2 (1 - \alpha\mu_g)^N + 4\frac{L^4}{\mu_g^2}D^2(1 - \alpha\mu_g)^N + 24(4p+15)\frac{L^2 M^2}{Q\mu_g^2}
$$

$$
+ \frac{\mu^2}{Q}L_{\mathcal{J}}^2 M^2 d\mathcal{P}_4(p) + \frac{24L^2 M^2(1 - \alpha\mu_g)^{2N}}{\mu_g^2} + 6\mu^2 L_{\mathcal{J}}^2 M^2 d(p+3)^3
$$

$$
+ \frac{48M^2(\tau\mu_g + L\rho)^2}{\mu_g^4}(1 - \alpha\mu_g)^{N-1}D^2
$$

$$
= \mathcal{D}_{var} = \mathcal{O}\left((1 - \alpha\mu_g)^N + \frac{p}{Q} + \frac{\mu^2 dp^4}{Q} + \mu^2 dp^3\right) \tag{28}
$$

*where the expectation $\mathbb{E}[\cdot]$ is conditioned on $x_k$ and $y_k^N$.*

*Proof of Proposition 2.* Based on the definitions of $\nabla\Phi(x_k)$ and $\widehat{\nabla}\Phi(x_k)$ and conditioning on $x_k$ and $y_k^N$, we have

$$
\mathbb{E}\big\|\widehat{\nabla}\Phi(x_k) - \nabla\Phi(x_k)\big\|^2
$$

$$
\leq 2\big\|\nabla_x f(x_k, y_k^N) - \nabla_x f(x_k, y_k^*)\big\|^2 + 2\mathbb{E}\big\|\hat{\mathcal{J}}_N^\top \nabla_y f(x_k, y_k^N) - \mathcal{J}_*^\top \nabla_y f(x_k, y_k^*)\big\|^2
$$

$$
\leq 2L^2\big\|y_k^N - y_k^*\big\|^2 + 4\big\|\mathcal{J}_*\big\|_F^2\big\|\nabla_y f(x_k, y_k^N) - \nabla_y f(x_k, y_k^*)\big\|^2
$$

$$
+ 4\mathbb{E}\big\|\hat{\mathcal{J}}_N - \mathcal{J}_*\big\|_F^2\big\|\nabla_y f(x_k, y_k^N)\big\|^2
$$

$$
\overset{(i)}{\leq} 2L^2 D^2(1 - \alpha\mu_g)^N + 4\frac{L^4}{\mu_g^2}\big\|y_k^N - y_k^*\big\|^2 + 4M^2\mathbb{E}\big\|\hat{\mathcal{J}}_N - \mathcal{J}_*\big\|_F^2
$$

$$
\overset{(ii)}{\leq} 2L^2 D^2(1 - \alpha\mu_g)^N + 4\frac{L^4}{\mu_g^2}D^2(1 - \alpha\mu_g)^N + 4M^2\mathbb{E}\big\|\hat{\mathcal{J}}_N - \mathcal{J}_*\big\|_F^2 \tag{29}
$$

where $(i)$ follows from Lemma 4 and Assumption 3, and $(i)$ and $(ii)$ also use the following result for full GD (when applied to a strongly-convex function).

$$
\big\|y_k^N - y_k^*\big\|^2 \leq (1 - \alpha\mu_g)^N D^2.
$$

Next, we upper-bound the last term $\mathbb{E}\big\|\hat{\mathcal{J}}_N - \mathcal{J}_*\big\|_F^2$ at the last line of eq. (29). First note that

$$
\mathbb{E}\big\|\hat{\mathcal{J}}_N - \mathcal{J}_*\big\|_F^2 \leq 3\mathbb{E}\big\|\hat{\mathcal{J}}_N - \mathcal{J}_\mu\big\|_F^2 + 3\big\|\mathcal{J}_N - \mathcal{J}_*\big\|_F^2 + 3\big\|\mathcal{J}_\mu - \mathcal{J}_N\big\|_F^2. \tag{30}
$$

We then upper-bound each term of the right hand side of eq. (30). For the first term, we have

$$
\mathbb{E}\big\|\hat{\mathcal{J}}_N - \mathcal{J}_\mu\big\|_F^2 = \mathbb{E}\big\|\frac{1}{Q}\sum_{j=1}^Q \hat{\mathcal{J}}_{N,j} - \mathcal{J}_\mu\big\|_F^2
$$

$$
= \frac{1}{Q^2}\mathbb{E}\big\|\sum_{j=1}^Q\left(\hat{\mathcal{J}}_{N,j} - \mathcal{J}_\mu\right)\big\|_F^2
$$

$$
= \frac{1}{Q^2}\mathbb{E}\left(\sum_{j=1}^Q\big\|\hat{\mathcal{J}}_{N,j} - \mathcal{J}_\mu\big\|_F^2 + 2\sum_{i<j}\left\langle\hat{\mathcal{J}}_{N,i} - \mathcal{J}_\mu, \hat{\mathcal{J}}_{N,j} - \mathcal{J}_\mu\right\rangle\right)
$$

$$
= \frac{1}{Q^2}\sum_{j=1}^Q\mathbb{E}\big\|\hat{\mathcal{J}}_{N,j} - \mathcal{J}_\mu\big\|_F^2
$$

$$
= \frac{1}{Q}\mathbb{E}\big\|\hat{\mathcal{J}}_{N,j} - \mathcal{J}_\mu\big\|_F^2, \quad j \in \{1, \ldots, Q\}. \tag{31}
$$

We next upper-bound the term $\mathbb{E}\big\|\hat{\mathcal{J}}_{N,j} - \mathcal{J}_\mu\big\|_F^2$ in eq. (31).

$$
\begin{aligned}
\mathbb{E}\big\|\hat{\mathcal{J}}_{N,j} - \mathcal{J}_\mu\big\|_F^2 &= \mathbb{E}\big\|\hat{\mathcal{J}}_{N,j}\big\|_F^2 - \big\|\mathcal{J}_\mu\big\|_F^2 \\
&\overset{(i)}{\leq} \sum_{i=1}^d \left( 4(p+4)\big\|\nabla y_{i,\mu}^N\big\|^2 + \frac{3}{2}\mu^2 L_{\mathcal{J}}^2 (p+5)^3 \right) - \sum_{i=1}^d \big\|\nabla y_{i,\mu}^N\big\|^2 \\
&\leq \sum_{i=1}^d \left( (4p+15)\big\|\nabla y_{i,\mu}^N\big\|^2 + \frac{3}{2}\mu^2 L_{\mathcal{J}}^2 (p+5)^3 \right),
\end{aligned}
\tag{32}
$$

where $(i)$ follows by applying Lemma 3 to the components of vector $y^N(x_k)$ which have Lipschitz gradients by Proposition 1. Then, noting that $\big\|\nabla y_{i,\mu}^N\big\|^2 \leq 2\big\|\nabla y_i^N\big\|^2 + \frac{1}{2}\mu^2 L_{\mathcal{J}}^2 (p+3)^3$ and replacing in eq. (32), we have

$$
\begin{aligned}
\mathbb{E}\big\|\hat{\mathcal{J}}_{N,j} - \mathcal{J}_\mu\big\|_F^2 &\leq \sum_{i=1}^d \left( 2(4p+15)\big\|\nabla y_i^N\big\|^2 + \mu^2 L_{\mathcal{J}}^2 \mathcal{P}_4(p) \right) \\
&\leq 2(4p+15)\big\|\mathcal{J}_N\big\|_F^2 + \mu^2 L_{\mathcal{J}}^2 d\mathcal{P}_4(p) \\
&\overset{(i)}{\leq} 2(4p+15)\frac{L^2}{\mu_g^2} + \mu^2 L_{\mathcal{J}}^2 d\mathcal{P}_4(p),
\end{aligned}
\tag{33}
$$

where $(i)$ follows from Lemma 5 and $\mathcal{P}_4$ is a polynomial of degree 4 in $p$. Combining eq. (31) and eq. (32) yields

$$
\mathbb{E}\big\|\hat{\mathcal{J}}_N - \mathcal{J}_\mu\big\|_F^2 \leq 2(4p+15)\frac{L^2}{Q\mu_g^2} + \frac{\mu^2}{Q} L_{\mathcal{J}}^2 d\mathcal{P}_4(p).
\tag{34}
$$

We next upper-bound the second term at the right hand side of eq. (30), which can be upper-bounded using eq. (41) in [28], as shown below.

$$
\big\|\mathcal{J}_N - \mathcal{J}_*\big\|^2 \leq \frac{2L^2(1-\alpha\mu_g)^{2N}}{\mu_g^2} + \frac{4(\tau\mu_g + L\rho)^2}{\mu_g^4}(1-\alpha\mu_g)^{N-1}D^2.
\tag{35}
$$

We finally upper-bound the last term at the right hand side of eq. (30) using Lemma 3.

$$
\begin{aligned}
\big\|\mathcal{J}_\mu - \mathcal{J}_N\big\|_F^2 &= \sum_{i=1}^d \big\|\nabla y_{i,\mu}^N - \nabla y_i^N\big\|^2 \\
&\leq \frac{\mu^2}{2} L_{\mathcal{J}}^2 d(p+3)^3.
\end{aligned}
\tag{36}
$$

Substituting eq. (34), eq. (35) and eq. (36) into eq. (30) yields

$$
\begin{aligned}
\mathbb{E}\big\|\hat{\mathcal{J}}_N - \mathcal{J}_*\big\|_F^2 \leq &\,6(4p+15)\frac{L^2}{Q\mu_g^2} + \frac{\mu^2}{Q} L_{\mathcal{J}}^2 d\mathcal{P}_4(p) + \frac{6L^2(1-\alpha\mu_g)^{2N}}{\mu_g^2} \\
&+ \frac{12(\tau\mu_g + L\rho)^2}{\mu_g^4}(1-\alpha\mu_g)^{N-1}D^2 + \frac{3\mu^2}{2} L_{\mathcal{J}}^2 d(p+3)^3.
\end{aligned}
\tag{37}
$$

Finally, the bound for the expected estimation error in eq. (29) becomes

$$
\begin{aligned}
\mathbb{E}\big\|\widehat{\nabla}\Phi(x_k) - \nabla\Phi(x_k)\big\|^2 \leq &\,2L^2 D^2(1-\alpha\mu_g)^N + 4\frac{L^4}{\mu_g^2}D^2(1-\alpha\mu_g)^N + 24(4p+15)\frac{L^2 M^2}{Q\mu_g^2} \\
&+ \frac{\mu^2}{Q} L_{\mathcal{J}}^2 M^2 d\mathcal{P}_4(p) + \frac{24L^2 M^2(1-\alpha\mu_g)^{2N}}{\mu_g^2} + 6\mu^2 L_{\mathcal{J}}^2 M^2 d(p+3)^3 \\
&+ \frac{48M^2(\tau\mu_g + L\rho)^2}{\mu_g^4}(1-\alpha\mu_g)^{N-1}D^2.
\end{aligned}
\tag{38}
$$

This completes the proof. $\qquad\square$

## I.2 Hypergradient Estimation Bias

**Lemma 8.** *Suppose that Assumptions 1, 2, and 3 hold. Then, the bias of hypergradient estimation can be upper-bounded as follows:*

$$\left\|\mathbb{E}\widehat{\nabla}\Phi(x_k) - \nabla\Phi(x_k)\right\| \leq \mathcal{D}_{bias} = \mathcal{O}\left((1-\alpha\mu_g)^{N/2} + \mu d^{1/2}p^{3/2}\right). \tag{39}$$

*Proof.* First note that We have

$$\begin{aligned}
\left\|\mathbb{E}\widehat{\nabla}\Phi(x_k) - \nabla\Phi(x_k)\right\| \\
&=\left\|\nabla_x f(x_k, y_k^N) - \nabla_x f(x_k, y_k^*)\right\| + \left\|\mathcal{J}_\mu^\top \nabla_y f(x_k, y_k^N) - \mathcal{J}_*^\top \nabla_y f(x_k, y_k^*)\right\| \\
&\leq L\left\|y_k^N - y_k^*\right\| + \left\|\mathcal{J}_\mu^\top \nabla_y f(x_k, y_k^N) - \mathcal{J}_*^\top \nabla_y f(x_k, y_k^N)\right\| \\
&\quad + \left\|\mathcal{J}_*^\top \nabla_y f(x_k, y_k^N) - \mathcal{J}_*^\top \nabla_y f(x_k, y_k^*)\right\| \\
&\leq L\left\|y_k^N - y_k^*\right\| + M\left\|\mathcal{J}_\mu - \mathcal{J}_*\right\| + \frac{L}{\mu_g}\left\|\nabla_y f(x_k, y_k^N) - \nabla_y f(x_k, y_k^*)\right\| \\
&\leq L\left\|y_k^N - y_k^*\right\| + M\left\|\mathcal{J}_\mu - \mathcal{J}_*\right\| + \frac{L^2}{\mu_g}\left\|y_k^N - y_k^*\right\|.
\end{aligned}$$

which, in conjunction with $\|\mathcal{J}_\mu - \mathcal{J}_*\| = \|\mathcal{J}_\mu - \mathcal{J}_N\| + \|\mathcal{J}_N - \mathcal{J}_*\| \leq \frac{\mu}{2}L_{\mathcal{J}}d^{1/2}(p+3)^{3/2} + \frac{L(1-\alpha\mu_g)^N}{\mu_g} + \frac{2(\tau\mu_g+L\rho)}{\mu_g^2}(1-\alpha\mu_g)^{(N-1)/2}D$, yields

$$\begin{aligned}
\left\|\mathbb{E}\widehat{\nabla}\Phi(x_k) - \nabla\Phi(x_k)\right\| \leq{}& LD(1-\alpha\mu_g)^{N/2} + \frac{\mu}{2}L_{\mathcal{J}}Md^{1/2}(p+3)^{3/2} + \frac{LM(1-\alpha\mu_g)^N}{\mu_g} \\
&+ \frac{2MD(\tau\mu_g+L\rho)}{\mu_g^2}(1-\alpha\mu_g)^{(N-1)/2} + \frac{L^2 D}{\mu_g}(1-\alpha\mu_g)^{N/2} \\
\leq{}& \mathcal{O}\left((1-\alpha\mu_g)^{N/2} + \mu d^{1/2}p^{3/2}\right).
\end{aligned}$$

Then, the proof is complete. $\qquad\square$

## I.3 Proof of Theorem 1

**Theorem 3** (Formal Statement of Theorem 1). *Suppose that Assumptions 1, 2, and 3 hold. Choose the inner- and outer-loop stepsizes respectively as $\alpha \leq \frac{1}{L}$ and $\beta = \frac{1}{L_\Phi\sqrt{K}}$, where $L_\Phi = L + \frac{2L^2+\tau M^2}{\mu_g} + \frac{\rho LM+L^3+\tau ML}{\mu_g^2} + \frac{\rho L^2 M}{\mu_g^3}$, and let $M_\Phi = \left(1 + \frac{L_g}{\mu_g}\right)M$. Further set $Q = \mathcal{O}(1)$ and $\mu = \mathcal{O}\left(\frac{1}{\sqrt{Kdp^3}}\right)$.
Then, the iterates $x_k$ for $k = 0, ..., K-1$ of PZOBO in Algorithm 1 satisfy:*

$$\frac{1-\frac{1}{\sqrt{K}}}{K}\sum_{k=0}^{K-1}\mathbb{E}\left\|\nabla\Phi(x_k)\right\|^2 \leq \frac{L_\phi(\Phi(x_0)-\Phi^*)}{\sqrt{K}} + M_\Phi \mathcal{D}_{bias} + \frac{\mathcal{D}_{var}}{\sqrt{K}} = \mathcal{O}\left(\frac{p}{\sqrt{K}} + (1-\alpha\mu_g)^N\right), \tag{40}$$

*with $\Phi^* = \inf_x \Phi(x)$, $\mathcal{D}_{bias}$ and $\mathcal{D}_{var}$ are defined in Lemma 8 and Proposition 6.*

*Proof of Theorem 1.* Using Assumptions 1, 2, and 3, we upper-bound the hypergradient $\nabla\Phi(x_k)$ by

$$\begin{aligned}
\left\|\nabla\Phi(x)\right\| ={}& \left\|\nabla_x f(x, y^*(x)) - \nabla_x \nabla_y g(x, y^*(x))\left[\nabla_y^2 g(x, y^*(x))\right]^{-1}\nabla_y f(x, y^*(x))\right\| \\
\leq{}& \left(1 + \frac{L_g}{\mu_g}\right)M. \tag{41}
\end{aligned}$$

Then, using the Lipschitzness of function $\Phi(x_k)$, we have

$$\begin{aligned}
\Phi(x_{k+1}) \leq{}& \Phi(x_k) + \langle\nabla\Phi(x_k), x_{k+1} - x_k\rangle + \frac{L_\phi}{2}\left\|x_{k+1} - x_k\right\|^2 \\
\leq{}& \Phi(x_k) - \beta\langle\nabla\Phi(x_k), \widehat{\nabla}\Phi(x_k)\rangle + \frac{L_\phi}{2}\beta^2\left\|\widehat{\nabla}\Phi(x_k)\right\|^2
\end{aligned}$$

$$\leq \Phi(x_k) - \beta\langle \nabla\Phi(x_k), \widehat{\nabla}\Phi(x_k) - \nabla\Phi(x_k)\rangle - \beta\|\nabla\Phi(x_k)\|^2$$
$$+ L_\phi\beta^2\left(\|\nabla\Phi(x_k)\|^2 + \|\widehat{\nabla}\Phi(x_k) - \nabla\Phi(x_k)\|^2\right) \tag{42}$$

Let $\mathbb{E}_k[\cdot] = \mathbb{E}_{u_{k,1:Q}}[\cdot|x_k, y_k^N]$ be the expectation over the Gaussian vectors $u_{k,1}, \ldots, u_{k,Q}$ conditioned on $x_k$ and $y_k^N$. Applying the expectation $\mathbb{E}_k[\cdot]$ to eq. (42) yields

$$\mathbb{E}_k\Phi(x_{k+1}) \leq \Phi(x_k) - \beta\langle \nabla\Phi(x_k), \mathbb{E}_k\widehat{\nabla}\Phi(x_k) - \nabla\Phi(x_k)\rangle - \beta\|\nabla\Phi(x_k)\|^2$$
$$+ L_\phi\beta^2\left(\|\nabla\Phi(x_k)\|^2 + \mathbb{E}_k\|\widehat{\nabla}\Phi(x_k) - \nabla\Phi(x_k)\|^2\right)$$
$$\leq \Phi(x_k) + \beta\|\nabla\Phi(x_k)\|\|\mathbb{E}_k\widehat{\nabla}\Phi(x_k) - \nabla\Phi(x_k)\| - (\beta - L_\phi\beta^2)\|\nabla\Phi(x_k)\|^2$$
$$+ L_\phi\beta^2\mathbb{E}_k\|\widehat{\nabla}\Phi(x_k) - \nabla\Phi(x_k)\|^2$$
$$\leq \Phi(x_k) + \beta M_\Phi \mathcal{D}_{bias} - (\beta - L_\phi\beta^2)\|\nabla\Phi(x_k)\|^2 + \beta^2 L_\phi\mathcal{D}_{var} \tag{43}$$

where $\mathcal{D}_{bias}$ and $\mathcal{D}_{var}$ represent respectively the upper-bound established for for the bias and variance. Now taking total expectation over $\mathcal{U}_k = \{u_{1,1:Q}, \ldots, u_{k,1:Q}\}$, we have

$$E_{k+1} \leq E_k - \beta(1 - L_\phi\beta)\mathbb{E}_{\mathcal{U}_k}\|\nabla\Phi(x_k)\|^2 + \beta M_\Phi\mathcal{D}_{bias} + \beta^2 L_\phi\mathcal{D}_{var} \tag{44}$$

where $E_k = \mathbb{E}_{\mathcal{U}_{k-1}}\Phi(x_k)$. Summing up the inequalities in eq. (44) for $k = 0, \ldots, K-1$ yields

$$E_K \leq E_0 - \beta(1 - L_\phi\beta)\sum_{k=0}^{K-1}\mathbb{E}_{\mathcal{U}_k}\|\nabla\Phi(x_k)\|^2 + \beta K M_\Phi\mathcal{D}_{bias} + \beta^2 K L_\phi\mathcal{D}_{var} \tag{45}$$

Setting $\beta = \frac{1}{L_\phi\sqrt{K}}$, denoting by $\Phi^* = \inf_x \Phi(x)$, and rearranging eq. (45), we have

$$\frac{1 - \frac{1}{\sqrt{K}}}{K}\sum_{k=0}^{K-1}\mathbb{E}_{\mathcal{U}_k}\|\nabla\Phi(x_k)\|^2 \leq \frac{L_\phi(\Phi(x_0) - \Phi^*)}{\sqrt{K}} + M_\Phi\mathcal{D}_{bias} + \frac{\mathcal{D}_{var}}{\sqrt{K}}.$$

Setting $Q = \mathcal{O}(1)$ and $\mu = \mathcal{O}\left(\frac{1}{\sqrt{K}dp^3}\right)$ in the expressions of $\mathcal{D}_{bias}$ and $\mathcal{D}_{var}$ finishes the proof. $\quad\square$

## J  Proofs for Stochastic Bilevel Optimization

Define the following quantities

$$\hat{\mathcal{J}}_{N,j} = \hat{\mathcal{J}}_N(x_k, u_j) = \begin{pmatrix} \frac{Y_1^N(x_k + \mu u_j; \mathcal{S}) - Y_1^N(x_k; \mathcal{S})}{\mu}u_j^\top \\ \vdots \\ \frac{Y_d^N(x_k + \mu u_j; \mathcal{S}) - Y_d^N(x_k; \mathcal{S})}{\mu}u_j^\top \end{pmatrix}, \quad \mathcal{J}_N = \frac{\partial Y_k^N}{\partial x_k}, \quad \mathcal{J}_* = \frac{\partial y_k^*}{\partial x_k}$$

where $u_j \in \mathbb{R}^p, j = 1, \ldots, Q$ are standard Gaussian vectors and $Y_k^N$ is the output of SGD obtained with the minibatches $\{\mathcal{S}_0, \ldots, \mathcal{S}_{N-1}\}$.

Conditioning on $x_k$ and $Y_k^N$ and taking expectation over $u_j$ yields

$$\mathbb{E}_{u_j}\hat{\mathcal{J}}_{N,j} = \mathbb{E}_{u_j}\begin{pmatrix} \frac{Y_1^N(x_k + \mu u_j; \mathcal{S}) - Y_1^N(x_k; \mathcal{S})}{\mu}u_j^\top \\ \vdots \\ \frac{Y_d^N(x_k + \mu u_j; \mathcal{S}) - Y_d^N(x_k; \mathcal{S})}{\mu}u_j^\top \end{pmatrix} = \begin{pmatrix} \nabla_x^\top Y_{1,\mu}^N(x_k; \mathcal{S}) \\ \vdots \\ \nabla_x^\top Y_{d,\mu}^N(x_k; \mathcal{S}) \end{pmatrix} = \mathcal{J}_\mu(\mathcal{S})$$

where $Y_{i,\mu}^N(x_k; \mathcal{S})$ is the $i$-th component of vector $Y_\mu^N(x_k; \mathcal{S})$, which is the entry-wise Gaussian smooth approximation of vector $Y^N(x_k; \mathcal{S})$. Let $\mathbb{E}_k[\cdot] = \mathbb{E}[\cdot|x_k, Y_k^N] = \mathbb{E}_{\mathcal{D}_F, u_{1:q}}$ be the expectation over the Gaussian vectors and the sample minibatch $\mathcal{D}_F$ conditioned on $x_k$ and $Y_k^N$.

### J.1 Proof of Proposition 3

**Proposition 7** (Formal Statement of Proposition 3). *Suppose that Assumptions 1, 2, and 4 hold. Choose the inner-loop stepsize as $\alpha = \frac{2}{L+\mu_g}$. Define the constants*

$$C_\gamma = (1 - \alpha\mu_g)\left(1 - \alpha\mu_g + \frac{\alpha}{\gamma} + \frac{\alpha L}{\gamma\mu_g}\right),$$

$$C_{xy} = \alpha\left(\alpha + \gamma(1 - \alpha\mu_g) + \alpha\frac{L}{\mu_g}\right), \quad C_y = \frac{L}{\mu_g}C_{xy}$$

$$\Gamma = 2(\tau^2 C_{xy} + \rho^2 C_y)\frac{\sigma^2}{\mu_g LS} + 2\frac{L^2}{S}(C_{xy} + C_y), \quad \lambda = 2(\tau^2 C_{xy} + \rho^2 C_y)D^2, \quad (46)$$

*where $\gamma$ is such that $\gamma \geq \frac{L+\mu_g}{\mu_g^2}$. Then, we have:*

$$\mathbb{E}\big\|\mathcal{J}_N - \mathcal{J}_*\big\|_F^2 \leq C_\gamma^N \frac{L^2}{\mu_g^2} + \frac{\lambda(L + \mu_g)^2(1 - \alpha\mu_g)C_\gamma^{N-1}}{(L + \mu_g)^2(1 - \alpha\mu_g) - (L - \mu_g)^2} + \frac{\Gamma}{1 - C_\gamma}.$$

*Proof of Proposition 7.* Based on the SGD updates, we have

$$Y_k^t = Y_k^{t-1} - \alpha\nabla_y G\left(x_k, Y_k^{t-1}; \mathcal{S}_{t-1}\right), \quad t = 1, \ldots, N.$$

Taking the derivatives w.r.t. $x_k$ yields

$$\mathcal{J}_t = \mathcal{J}_{t-1} - \alpha\nabla_x\nabla_y G\left(x_k, Y_k^{t-1}; \mathcal{S}_{t-1}\right) - \alpha\mathcal{J}_{t-1}\nabla_y^2 G\left(x_k, Y_k^{t-1}; \mathcal{S}_{t-1}\right),$$

which further yields

$$\begin{aligned}
\mathcal{J}_t - \mathcal{J}_* ={}& \mathcal{J}_{t-1} - \mathcal{J}_* - \alpha\nabla_x\nabla_y G\left(x_k, Y_k^{t-1}; \mathcal{S}_{t-1}\right) - \alpha\mathcal{J}_{t-1}\nabla_y^2 G\left(x_k, Y_k^{t-1}; \mathcal{S}_{t-1}\right) \\
& + \alpha\left(\nabla_x\nabla_y g\left(x_k, y_k^*\right) + \mathcal{J}_*\nabla_y^2 g\left(x_k, y_k^*\right)\right) \\
={}& \mathcal{J}_{t-1} - \mathcal{J}_* - \alpha\left(\nabla_x\nabla_y G\left(x_k, Y_k^{t-1}; \mathcal{S}_{t-1}\right) - \nabla_x\nabla_y g\left(x_k, y_k^*\right)\right) \\
& - \alpha\left(\mathcal{J}_{t-1} - \mathcal{J}_*\right)\nabla_y^2 G\left(x_k, Y_k^{t-1}; \mathcal{S}_{t-1}\right) \\
& + \alpha\mathcal{J}_*\left(\nabla_y^2 g\left(x_k, y_k^*\right) - \nabla_y^2 G\left(x_k, Y_k^{t-1}; \mathcal{S}_{t-1}\right)\right).
\end{aligned}$$

Hence, using the triangle inequality, we have

$$\begin{aligned}
\big\|\mathcal{J}_t - \mathcal{J}_*\big\|_F \overset{(i)}{\leq}{}& \big\|\left(\mathcal{J}_{t-1} - \mathcal{J}_*\right)\left(I - \nabla_y^2 G\left(x_k, Y_k^{t-1}; \mathcal{S}_{t-1}\right)\right)\big\|_F \\
& + \alpha\big\|\nabla_x\nabla_y G\left(x_k, Y_k^{t-1}; \mathcal{S}_{t-1}\right) - \nabla_x\nabla_y g\left(x_k, y_k^*\right)\big\|_F \\
& + \alpha\big\|\mathcal{J}_*\left(\nabla_y^2 G\left(x_k, Y_k^{t-1}; \mathcal{S}_{t-1}\right) - \nabla_y^2 g\left(x_k, y_k^*\right)\right)\big\|_F,
\end{aligned}$$

where $(i)$ follows from Assumption 1. We then further have

$$\begin{aligned}
&\big\|\mathcal{J}_t - \mathcal{J}_*\big\|_F^2 \\
&\leq (1 - \alpha\mu_g)^2\big\|\mathcal{J}_{t-1} - \mathcal{J}_*\big\|_F^2 + \alpha^2\big\|\nabla_x\nabla_y G\left(x_k, Y_k^{t-1}; \mathcal{S}_{t-1}\right) - \nabla_x\nabla_y g\left(x_k, y_k^*\right)\big\|_F^2 \\
&\quad + \alpha^2\frac{L^2}{\mu_g^2}\big\|\nabla_y^2 G\left(x_k, Y_k^{t-1}; \mathcal{S}_{t-1}\right) - \nabla_y^2 g\left(x_k, y_k^*\right)\big\|_F^2 \\
&\quad + 2\alpha(1 - \alpha\mu_g)\underbrace{\big\|\mathcal{J}_{t-1} - \mathcal{J}_*\big\|_F\big\|\nabla_x\nabla_y G\left(x_k, Y_k^{t-1}; \mathcal{S}_{t-1}\right) - \nabla_x\nabla_y g\left(x_k, y_k^*\right)\big\|_F}_{P_1} \\
&\quad + 2\alpha(1 - \alpha\mu_g)\frac{L}{\mu_g}\underbrace{\big\|\mathcal{J}_{t-1} - \mathcal{J}_*\big\|_F\big\|\nabla_y^2 G\left(x_k, Y_k^{t-1}; \mathcal{S}_{t-1}\right) - \nabla_y^2 g\left(x_k, y_k^*\right)\big\|_F}_{P_2} \\
&\quad + 2\alpha^2\frac{L}{\mu_g}\underbrace{\big\|\nabla_y^2 G\left(x_k, Y_k^{t-1}; \mathcal{S}_{t-1}\right) - \nabla_y^2 g\left(x_k, y_k^*\right)\big\|_F\big\|\nabla_x\nabla_y G\left(x_k, Y_k^{t-1}; \mathcal{S}_{t-1}\right) - \nabla_x\nabla_y g\left(x_k, y_k^*\right)\big\|_F}_{P_3}.
\end{aligned}$$

The terms $P_1$, $P_2$ and $P_3$ in the above inequality can be transformed as follows using the Peter-Paul version of Young's inequality.

$$P_1 \leq \frac{1}{2\gamma}\big\|\mathcal{J}_{t-1} - \mathcal{J}_*\big\|_F^2 + \frac{\gamma}{2}\big\|\nabla_x\nabla_y G\left(x_k, Y_k^{t-1}; \mathcal{S}_{t-1}\right) - \nabla_x\nabla_y g\left(x_k, y_k^*\right)\big\|_F^2, \quad \gamma > 0$$

$$P_2 \leq \frac{1}{2\gamma}\big\|\mathcal{J}_{t-1} - \mathcal{J}_*\big\|_F^2 + \frac{\gamma}{2}\big\|\nabla_y^2 G\left(x_k, Y_k^{t-1}; \mathcal{S}_{t-1}\right) - \nabla_y^2 g\left(x_k, y_k^*\right)\big\|_F^2, \quad \gamma > 0$$

$$P_3 \leq \frac{1}{2}\big\|\nabla_y^2 G\left(x_k, Y_k^{t-1}; \mathcal{S}_{t-1}\right) - \nabla_y^2 g\left(x_k, y_k^*\right)\big\|_F^2$$
$$+ \frac{1}{2}\big\|\nabla_x \nabla_y G\left(x_k, Y_k^{t-1}; \mathcal{S}_{t-1}\right) - \nabla_x \nabla_y g\left(x_k, y_k^*\right)\big\|_F^2.$$

Note that the trade-off constant $\gamma$ controls the contraction coefficient (i.e., the factor in front of $\big\|\mathcal{J}_{t-1} - \mathcal{J}_*\big\|_F^2$). Hence, we have

$$\big\|\mathcal{J}_t - \mathcal{J}_*\big\|_F^2$$
$$\leq \left( (1-\alpha\mu_g)^2 + \frac{\alpha}{\gamma}(1-\alpha\mu_g) + \frac{\alpha L}{\gamma\mu_g}(1-\alpha\mu_g) \right)\big\|\mathcal{J}_{t-1} - \mathcal{J}_*\big\|_F^2$$
$$+ \left( \alpha^2 + \alpha\gamma(1-\alpha\mu_g) + \alpha^2 \frac{L}{\mu_g} \right)\big\|\nabla_x \nabla_y G\left(x_k, Y_k^{t-1}; \mathcal{S}_{t-1}\right) - \nabla_x \nabla_y g\left(x_k, y_k^*\right)\big\|_F^2$$
$$+ \left( \alpha^2 \frac{L^2}{\mu_g^2} + \alpha\gamma\frac{L}{\mu_g}(1-\alpha\mu_g) + \alpha^2\frac{L}{\mu_g} \right)\big\|\nabla_y^2 G\left(x_k, Y_k^{t-1}; \mathcal{S}_{t-1}\right) - \nabla_y^2 g\left(x_k, y_k^*\right)\big\|_F^2.$$

Let $\mathbb{E}_{t-1}[\cdot] = \mathbb{E}[\cdot | x_k, Y_k^{t-1}]$. Conditioning on $x_k$ and $Y_k^{t-1}$ and taking expectations yield

$$\mathbb{E}_{t-1}\big\|\mathcal{J}_t - \mathcal{J}_*\big\|_F^2$$
$$\leq C_\gamma\big\|\mathcal{J}_{t-1} - \mathcal{J}_*\big\|_F^2 + C_{xy}\mathbb{E}_{t-1}\big\|\nabla_x \nabla_y G\left(x_k, Y_k^{t-1}; \mathcal{S}_{t-1}\right) - \nabla_x \nabla_y g\left(x_k, y_k^*\right)\big\|_F^2$$
$$+ C_y\mathbb{E}_{t-1}\big\|\nabla_y^2 G\left(x_k, Y_k^{t-1}; \mathcal{S}_{t-1}\right) - \nabla_y^2 g\left(x_k, y_k^*\right)\big\|_F^2, \tag{47}$$

where $C_\gamma$, $C_{xy}$ and $C_y$ are defined as follows

$$C_\gamma = (1-\alpha\mu_g)\left( 1-\alpha\mu_g + \frac{\alpha}{\gamma} + \frac{\alpha L}{\gamma\mu_g} \right), C_{xy} = \alpha\left( \alpha + \gamma(1-\alpha\mu_g) + \alpha\frac{L}{\mu_g} \right), C_y = \frac{L}{\mu_g}C_{xy}.$$

Conditioning on $x_k$ and $Y_k^{t-1}$, we have

$$\mathbb{E}_{t-1}\big\|\nabla_x \nabla_y G\left(x_k, Y_k^{t-1}; \mathcal{S}_{t-1}\right) - \nabla_x \nabla_y g\left(x_k, y_k^*\right)\big\|_F^2$$
$$\leq 2\mathbb{E}_{t-1}\big\|\nabla_x \nabla_y g\left(x_k, Y_k^{t-1}\right) - \nabla_x \nabla_y g\left(x_k, y_k^*\right)\big\|_F^2$$
$$+ 2\mathbb{E}_{t-1}\big\|\nabla_x \nabla_y G\left(x_k, Y_k^{t-1}; \mathcal{S}_{t-1}\right) - \nabla_x \nabla_y g\left(x_k, Y_k^{t-1}\right)\big\|_F^2$$
$$\overset{(i)}{\leq} 2\frac{L^2}{S} + 2\tau^2\big\|Y_k^{t-1} - y_k^*\big\|^2, \tag{48}$$

where $(i)$ follows from Lemma 1 and Assumption 2. Similarly we can derive

$$\mathbb{E}_{t-1}\big\|\nabla_y^2 G\left(x_k, Y_k^{t-1}; \mathcal{S}_{t-1}\right) - \nabla_y^2 g\left(x_k, y_k^*\right)\big\|_F^2 \leq 2\frac{L^2}{S} + 2\rho^2\big\|Y_k^{t-1} - y_k^*\big\|^2. \tag{49}$$

Combining eq. (47), eq. (48), and eq. (49) we obtain

$$\mathbb{E}_{t-1}\big\|\mathcal{J}_t - \mathcal{J}_*\big\|_F^2 \leq C_\gamma\big\|\mathcal{J}_{t-1} - \mathcal{J}_*\big\|_F^2 + 2(\tau^2 C_{xy} + \rho^2 C_y)\big\|Y_k^{t-1} - y_k^*\big\|^2$$
$$+ 2\frac{L^2}{S}(C_{xy} + C_y). \tag{50}$$

Unconditioning on $x_k$ and $Y_k^{t-1}$ and taking total expectations of eq. (50) yield

$$\mathbb{E}\big\|\mathcal{J}_t - \mathcal{J}_*\big\|_F^2 \leq C_\gamma\mathbb{E}\big\|\mathcal{J}_{t-1} - \mathcal{J}_*\big\|_F^2 + 2(\tau^2 C_{xy} + \rho^2 C_y)\mathbb{E}\big\|Y_k^{t-1} - y_k^*\big\|^2 + 2\frac{L^2}{S}(C_{xy} + C_y)$$
$$\overset{(i)}{\leq} C_\gamma\mathbb{E}\big\|\mathcal{J}_{t-1} - \mathcal{J}_*\big\|_F^2 + 2(\tau^2 C_{xy} + \rho^2 C_y)\left( \left(\frac{L-\mu_g}{L+\mu_g}\right)^{2(t-1)} D^2 + \frac{\sigma^2}{\mu_g LS} \right)$$
$$+ 2\frac{L^2}{S}(C_{xy} + C_y),$$

where $(i)$ follows from the analysis of SGD for a strongly-convex function. Let $\Gamma = 2(\tau^2 C_{xy} + \rho^2 C_y)\frac{\sigma^2}{\mu_g LS} + 2\frac{L^2}{S}(C_{xy} + C_y)$ and $\lambda = 2(\tau^2 C_{xy} + \rho^2 C_y)D^2$. Then, we have

$$\mathbb{E}\|\mathcal{J}_t - \mathcal{J}_*\|_F^2 \leq C_\gamma \mathbb{E}\|\mathcal{J}_{t-1} - \mathcal{J}_*\|_F^2 + \lambda\left(\frac{L-\mu_g}{L+\mu_g}\right)^{2(t-1)} + \Gamma. \tag{51}$$

Telescoping eq. (51) over $t$ from $N$ down to 1 yields

$$\mathbb{E}\|\mathcal{J}_N - \mathcal{J}_*\|_F^2 \leq C_\gamma^N \mathbb{E}\|\mathcal{J}_0 - \mathcal{J}_*\|_F^2 + \lambda \sum_{t=0}^{N-1}\left(\frac{L-\mu_g}{L+\mu_g}\right)^{2t} C_\gamma^{N-1-t} + \Gamma \sum_{t=0}^{N-1} C_\gamma^t$$

which, in conjunction with $\left(\frac{L-\mu_g}{L+\mu_g}\right)^2 \leq 1 - \alpha\mu_g$ and $\gamma \geq \frac{L+\mu_g}{\mu_g^2}$ such that $C_\gamma \leq 1 - \alpha\mu_g$, yields

$$\mathbb{E}\|\mathcal{J}_N - \mathcal{J}_*\|_F^2 \leq C_\gamma^N \frac{L^2}{\mu_g^2} + \lambda C_\gamma^{N-1} \sum_{t=0}^{N-1}\left(\frac{(L-\mu_g)^2}{(L+\mu_g)^2(1-\alpha\mu_g)}\right)^t + \frac{\Gamma}{1-C_\gamma}$$

$$\leq C_\gamma^N \frac{L^2}{\mu_g^2} + \frac{\lambda(L+\mu_g)^2(1-\alpha\mu_g)C_\gamma^{N-1}}{(L+\mu_g)^2(1-\alpha\mu_g) - (L-\mu_g)^2} + \frac{\Gamma}{1-C_\gamma}. \tag{52}$$

The proof is then completed. $\qquad\square$

**Lemma 9.** *Suppose that Assumptions 1, 2, 3, and 4 hold. Set the inner-loop stepsize as $\alpha = \frac{2}{L+\mu_g}$. Then, we have*

$$\mathbb{E}\|\mathbb{E}_k\widehat{\nabla}\Phi(x_k) - \nabla\Phi(x_k)\|^2$$

$$\leq 8M^2\left(C_\gamma^N \frac{L^2}{\mu_g^2} + \frac{\lambda(L+\mu_g)^2(1-\alpha\mu_g)C_\gamma^{N-1}}{(L+\mu_g)^2(1-\alpha\mu_g) - (L-\mu_g)^2} + \frac{\Gamma}{1-C_\gamma} + \frac{\mu^2}{2}L_{\mathcal{J}}^2 d(p+3)^3\right)$$

$$+ 2L^2\left(1 + 2\frac{L^2}{\mu_g^2}\right)\left(\left(\frac{L-\mu_g}{L+\mu_g}\right)^{2N} D^2 + \frac{\sigma^2}{\mu_g LS}\right), \tag{53}$$

*where the expectation $\mathbb{E}_k[\cdot]$ is conditioned on $x_k$ and $Y_k^N$.*

*Proof of Lemma 9.* Conditioning on $x_k$ and $Y_k^N$, we have

$$\mathbb{E}_k\widehat{\nabla}\Phi(x_k) = \nabla_x f(x_k, Y_k^N) + \mathcal{J}_\mu^\top \nabla_y f(x_k, Y_k^N).$$

Recall $\nabla\Phi(x_k) = \nabla_x f(x_k, y_k^*) + \mathcal{J}_*^\top \nabla_y f(x_k, y_k^*)$. Thus, we have

$$\left\|\mathbb{E}_k\widehat{\nabla}\Phi(x_k) - \nabla\Phi(x_k)\right\|^2$$

$$\leq 2\left\|\nabla_x f(x_k, Y_k^N) - \nabla_x f(x_k, y_k^*)\right\|^2 + 2\left\|\mathcal{J}_\mu^\top \nabla_y f(x_k, Y_k^N) - \mathcal{J}_*^\top \nabla_y f(x_k, y_k^*)\right\|^2$$

$$\leq 2L^2\left\|Y_k^N - y_k^*\right\|^2 + 4\left\|\mathcal{J}_\mu^\top \nabla_y f(x_k, Y_k^N) - \mathcal{J}_*^\top \nabla_y f(x_k, Y_k^N)\right\|^2$$

$$+ 4\left\|\mathcal{J}_*^\top \nabla_y f(x_k, Y_k^N) - \mathcal{J}_*^\top \nabla_y f(x_k, y_k^*)\right\|^2$$

$$\overset{(i)}{\leq} 2L^2\left\|Y_k^N - y_k^*\right\|^2 + 4M^2\left\|\mathcal{J}_\mu - \mathcal{J}_*\right\|_F^2 + 4\frac{L^2}{\mu_g^2}\left\|\nabla_y f(x_k, Y_k^N) - \nabla_y f(x_k, y_k^*)\right\|^2$$

$$\leq 2L^2\left\|Y_k^N - y_k^*\right\|^2 + 8M^2\left\|\mathcal{J}_\mu - \mathcal{J}_N\right\|_F^2 + 8M^2\left\|\mathcal{J}_N - \mathcal{J}_*\right\|_F^2 + 4\frac{L^4}{\mu_g^2}\left\|Y_k^N - y_k^*\right\|^2,$$

where $(i)$ applies Lemma 4 and Assumption 3. Taking expectation of the above inequality yields

$$\mathbb{E}\|\mathbb{E}_k\widehat{\nabla}\Phi(x_k) - \nabla\Phi(x_k)\|^2 \leq 2L^2\left(1 + 2\frac{L^2}{\mu_g^2}\right)\mathbb{E}\|Y_k^N - y_k^*\|^2 + 8M^2\mathbb{E}\|\mathcal{J}_N - \mathcal{J}_*\|_F^2$$

$$+ 8M^2\mathbb{E}\|\mathcal{J}_\mu - \mathcal{J}_N\|_F^2. \tag{54}$$

Using the fact that $Y_i^N(x_k; \cdot)$ has Lipschitz gradient (see Proposition 1) and Lemma 3, the last term at the right hand side of eq. (54) can be directly upper-bounded as

$$\left\|\mathcal{J}_\mu - \mathcal{J}_N\right\|_F^2 = \sum_{i=1}^d \left\|\nabla Y_{i,\mu}^N(x_k; \mathcal{S}) - \nabla Y_i^N(x_k; \mathcal{S})\right\|^2 \leq \frac{\mu^2}{2} L_{\mathcal{J}}^2 d(p+3)^3, \qquad (55)$$

where $L_{\mathcal{J}}$ is the Lipschitz constant of the Jacobian $\mathcal{J}_N$ (and also of its rows $\nabla Y_i^N(x_k; \mathcal{S})$) as defined in Proposition 1. Combining eq. (54), eq. (55), Proposition 7, and SGD analysis (as in eq. (60) in [28]) yields

$$\mathbb{E}\left\|\mathbb{E}_k \widehat{\nabla} \Phi(x_k) - \nabla \Phi(x_k)\right\|^2$$
$$\leq 8M^2 \left(C_\gamma^N \frac{L^2}{\mu_g^2} + \frac{\lambda(L+\mu_g)^2(1-\alpha\mu_g)C_\gamma^{N-1}}{(L+\mu_g)^2(1-\alpha\mu_g) - (L-\mu_g)^2} + \frac{\Gamma}{1-C_\gamma} + \frac{\mu^2}{2} L_{\mathcal{J}}^2 d(p+3)^3\right)$$
$$+ 2L^2 \left(1 + 2\frac{L^2}{\mu_g^2}\right) \left(\left(\frac{L-\mu_g}{L+\mu_g}\right)^{2N} D^2 + \frac{\sigma^2}{\mu_g LS}\right). \qquad (56)$$

This finishes the proof. □

## J.2 Proof of Proposition 4

**Proposition 8** (Formal Statement of Proposition 4). *Suppose that Assumptions 1, 2, 3, and 4 hold. Set the inner-loop stepsize as $\alpha = \frac{2}{L+\mu_g}$. Then, we have:*

$$\mathbb{E}\left\|\widehat{\nabla}\Phi(x_k) - \nabla\Phi(x_k)\right\|^2 \leq \Delta + \mathcal{B}_1 \qquad (57)$$

*where $\Delta = 8M^2 \left(\left(1 + \frac{1}{D_f}\right)\frac{4p+15}{Q} + \frac{1}{D_f}\right)\frac{L^2}{\mu_g^2} + 2\frac{M^2}{D_f} + \left(1 + \frac{1}{D_f}\right)\frac{4M^2}{Q}\mu^2 dL_{\mathcal{J}}^2 \mathcal{P}_4(p) + \frac{4M^2}{D_f}\mu^2 dL_{\mathcal{J}}^2 \mathcal{P}_3(p)$ and $\mathcal{B}_1$ respresents the upper bound established in Lemma 9.*

*Proof of Proposition 8.* We have, conditioning on $x_k$ and $Y_k^N$

$$\mathbb{E}_k\left\|\widehat{\nabla}\Phi(x_k) - \nabla\Phi(x_k)\right\|^2 = \mathbb{E}_k\left\|\widehat{\nabla}\Phi(x_k) - \mathbb{E}_k\widehat{\nabla}\Phi(x_k)\right\|^2 + \left\|\mathbb{E}_k\widehat{\nabla}\Phi(x_k) - \nabla\Phi(x_k)\right\|^2. \qquad (58)$$

Our next step is to upper-bound the first term in eq. (58).

$$\mathbb{E}_k\left\|\widehat{\nabla}\Phi(x_k) - \mathbb{E}_k\widehat{\nabla}\Phi(x_k)\right\|^2$$
$$\leq 2\mathbb{E}_k\left\|\nabla_x F(x_k, Y_k^N; \mathcal{D}_F) - \nabla_x f(x_k, Y_k^N)\right\|^2$$
$$+ 2\mathbb{E}_k\left\|\hat{\mathcal{J}}_N^\top \nabla_y F(x_k, Y_k^N; \mathcal{D}_F) - \mathcal{J}_\mu^\top \nabla_y f(x_k, Y_k^N)\right\|^2$$
$$\leq 2\frac{M^2}{D_f} + 4\mathbb{E}_k\left\|\nabla_y F(x_k, Y_k^N; \mathcal{D}_F)\right\|^2 \left\|\hat{\mathcal{J}}_N - \mathcal{J}_\mu\right\|_F^2$$
$$+ 4\mathbb{E}_k\left\|\mathcal{J}_\mu\right\|_F^2 \left\|\nabla_y F(x_k, Y_k^N; \mathcal{D}_F) - \nabla_y f(x_k, Y_k^N)\right\|^2$$
$$\leq 2\frac{M^2}{D_f} + 4M^2 \left(1 + \frac{1}{D_f}\right)\mathbb{E}_k\left\|\hat{\mathcal{J}}_N - \mathcal{J}_\mu\right\|_F^2 + 4\frac{M^2}{D_f}\left\|\mathcal{J}_\mu\right\|_F^2, \qquad (59)$$

where the last two steps follow from Lemma 1.

Next, we upper-bound the term $\mathbb{E}_k\left\|\hat{\mathcal{J}}_N - \mathcal{J}_\mu\right\|_F^2$.

$$\mathbb{E}_k\left\|\hat{\mathcal{J}}_N - \mathcal{J}_\mu\right\|_F^2 = \frac{1}{Q}\mathbb{E}_k\left\|\hat{\mathcal{J}}_{N,j} - \mathcal{J}_\mu\right\|_F^2, \quad j \in \{1, \ldots, Q\}$$
$$\leq \frac{1}{Q}\left(\mathbb{E}_k\left\|\hat{\mathcal{J}}_{N,j}\right\|_F^2 - \left\|\mathcal{J}_\mu\right\|_F^2\right)$$
$$\leq \frac{1}{Q}\sum_{i=1}^d \left(\mathbb{E}_k\left\|\frac{Y_i^N(x_k + \mu u_j; \mathcal{S}) - Y_i^N(x_k; \mathcal{S})}{\mu}u_j\right\|^2 - \left\|\nabla Y_{i,\mu}^N(x_k; \mathcal{S})\right\|^2\right). \qquad (60)$$

Recall that for a function $h$ with $L$-Lipschitz gradient, we have

$$\mathbb{E}_u \left\| \frac{h(x+\mu u) - h(x)}{\mu} u \right\|^2 \le 4(p+4)\left\| \nabla h_\mu(x) \right\|^2 + \frac{3}{2}\mu^2 L^2 (p+5)^3. \tag{61}$$

Then, applying eq. (61) to function $Y_i^N(\cdot; \mathcal{S})$ yields

$$\mathbb{E}_{u_j} \left\| \frac{Y_i^N(x_k + \mu u_j; \mathcal{S}) - Y_i^N(x_k; \mathcal{S})}{\mu} u_j \right\|^2$$

$$\le 4(p+4)\left\| \nabla Y_{i,\mu}^N(x_k; \mathcal{S}) \right\|^2 + \frac{3}{2}\mu^2 L_{\mathcal{J}}^2 (p+5)^3.$$

Hence, eq. (60) becomes

$$\mathbb{E}_k \left\| \hat{\mathcal{J}}_N - \mathcal{J}_\mu \right\|^2 \le \frac{4p+15}{Q} \sum_{i=1}^d \left\| \nabla Y_{i,\mu}^N(x_k; \mathcal{S}) \right\|^2 + \frac{3\mu^2 dL_{\mathcal{J}}^2}{2Q}(p+5)^3$$

$$\le \frac{2(4p+15)}{Q} \sum_{i=1}^d \left\| \nabla Y_i^N(x_k; \mathcal{S}) \right\|^2 + \frac{\mu^2 dL_{\mathcal{J}}^2}{Q}\mathcal{P}_4(p)$$

$$\le \frac{2(4p+15)}{Q} \left\| \mathcal{J}_N \right\|_F^2 + \frac{\mu^2 dL_{\mathcal{J}}^2}{Q}\mathcal{P}_4(p), \tag{62}$$

where $\mathcal{P}_4(\cdot)$ is a polynomial of degree 4 in $p$. Combining eq. (59), eq. (62) and Lemma 3 yields

$$\mathbb{E}_k \left\| \widehat{\nabla}\Phi(x_k) - \mathbb{E}_k \widehat{\nabla}\Phi(x_k) \right\|^2 \le 2\frac{M^2}{D_f} + 4M^2\left(1 + \frac{1}{D_f}\right)\left(\frac{2(4p+15)}{Q}\left\|\mathcal{J}_N\right\|_F^2 + \frac{\mu^2 dL_{\mathcal{J}}^2}{Q}\mathcal{P}_4(p)\right)$$

$$+ \frac{4M^2}{D_f}\left(2\left\|\mathcal{J}_N\right\|_F^2 + \mu^2 dL_{\mathcal{J}}^2 \mathcal{P}_3(p)\right)$$

$$\le 8M^2\left(\left(1 + \frac{1}{D_f}\right)\frac{4p+15}{Q} + \frac{1}{D_f}\right)\frac{L^2}{\mu_g^2} + 2\frac{M^2}{D_f}$$

$$+ \left(1 + \frac{1}{D_f}\right)\frac{4M^2}{Q}\mu^2 dL_{\mathcal{J}}^2 \mathcal{P}_4(p) + \frac{4M^2}{D_f}\mu^2 dL_{\mathcal{J}}^2 \mathcal{P}_3(p)$$

$$= \Delta, \tag{63}$$

where $\Delta = 8M^2\left(\left(1 + \frac{1}{D_f}\right)\frac{4p+15}{Q} + \frac{1}{D_f}\right)\frac{L^2}{\mu_g^2} + 2\frac{M^2}{D_f} + \left(1 + \frac{1}{D_f}\right)\frac{4M^2}{Q}\mu^2 dL_{\mathcal{J}}^2 \mathcal{P}_4(p) + \frac{4M^2}{D_f}\mu^2 dL_{\mathcal{J}}^2 \mathcal{P}_3(p)$.

Taking total expectations on both eq. (58) and eq. (63) and combining the resulting equations, we have

$$\mathbb{E}\left\| \widehat{\nabla}\Phi(x_k) - \nabla\Phi(x_k) \right\|^2 = \mathbb{E}\left\| \widehat{\nabla}\Phi(x_k) - \mathbb{E}_k\widehat{\nabla}\Phi(x_k) \right\|^2 + \mathbb{E}\left\| \mathbb{E}_k\widehat{\nabla}\Phi(x_k) - \nabla\Phi(x_k) \right\|^2$$

$$\le \Delta + \mathcal{B}_1 \tag{64}$$

where $\mathcal{B}_1$ represents the upper-bound established in eq. (56). This then completes the proof. $\qquad\square$

## J.3   Proof of Theorem 2

**Theorem 4** (Formal Statement of Theorem 2). *Suppose that Assumptions 1, 2, 3, and 4 hold. Set the inner- and outer-loop stepsizes respectivelly as $\alpha = \frac{2}{L+\mu_g}$ and $\beta = \frac{1}{L_\Phi \sqrt{K}}$, where $L = \max\{L_f, L_g\}$ and the constants $L_\Phi$ and $M_\Phi$ are defined as in Theorem 3. Further, set $Q = \mathcal{O}(1)$, $D_f = \mathcal{O}(1)$, and $\mu = \mathcal{O}\left(\frac{1}{\sqrt{K}dp^3}\right)$. Then, the iterates $x_k, k = 0, ..., K-1$ of the PZOBO-S algorithm satisfy*

$$\frac{1 - \frac{1}{\sqrt{K}}}{K} \sum_{k=0}^{K-1} \mathbb{E}\left\| \nabla\Phi(x_k) \right\|^2$$

$$\le \frac{(\Phi(x_0) - \Phi^*)L_\Phi}{\sqrt{K}} + M_\Phi\sqrt{\mathcal{B}_1} + \frac{\mathcal{B}_1 + \Delta}{\sqrt{K}} = \mathcal{O}\left(\frac{p}{\sqrt{K}} + (1 - \alpha\mu_g)^N + \frac{1}{\sqrt{S}}\right), \tag{65}$$

*where $\Delta$ and $\mathcal{B}_1$ are given by*

$$\Delta = 8M^2 \left( \left(1 + \frac{1}{D_f}\right) \frac{4p+15}{Q} + \frac{1}{D_f} \right) \frac{L^2}{\mu_g^2} + 2\frac{M^2}{D_f} + \left(1 + \frac{1}{D_f}\right) \frac{4M^2}{Q} \mu^2 dL_{\mathcal{J}}^2 \mathcal{P}_4(p)$$

$$+ \frac{4M^2}{D_f} \mu^2 dL_{\mathcal{J}}^2 \mathcal{P}_3(p)$$

$$\mathcal{B}_1 = 8M^2 \left( C_\gamma^N \frac{L^2}{\mu_g^2} + \frac{\lambda(L+\mu_g)^2(1-\alpha\mu_g)C_\gamma^{N-1}}{(L+\mu_g)^2(1-\alpha\mu_g) - (L-\mu_g)^2} + \frac{\Gamma}{1-C_\gamma} + \frac{\mu^2}{2} L_{\mathcal{J}}^2 d(p+3)^3 \right)$$

$$+ 2L^2 \left(1 + 2\frac{L^2}{\mu_g^2}\right) \left( \left(\frac{L-\mu_g}{L+\mu_g}\right)^{2N} D^2 + \frac{\sigma^2}{\mu_g LS} \right)$$

*and the constants $\Gamma$, $\lambda$, and $C_\gamma$ are defined in Proposition 7.*

*Proof of Theorem 2.* Using the Lipschitzness of function $\Phi(x_k)$, we have

$$\Phi(x_{k+1}) \leq \Phi(x_k) + \langle \nabla\Phi(x_k), x_{k+1} - x_k \rangle + \frac{L_\phi}{2} \|x_{k+1} - x_k\|^2$$

$$\leq \Phi(x_k) - \beta\langle \nabla\Phi(x_k), \widehat{\nabla}\Phi(x_k) \rangle + \frac{L_\phi}{2}\beta^2 \|\widehat{\nabla}\Phi(x_k) - \nabla\Phi(x_k) + \nabla\Phi(x_k)\|^2$$

$$\leq \Phi(x_k) - \beta\langle \nabla\Phi(x_k), \widehat{\nabla}\Phi(x_k) \rangle + L_\phi\beta^2 \left( \|\nabla\Phi(x_k)\|^2 + \|\widehat{\nabla}\Phi(x_k) - \nabla\Phi(x_k)\|^2 \right)$$

Hence, taking expectation over the above inequality yields

$$\mathbb{E}\Phi(x_{k+1}) \leq \mathbb{E}\Phi(x_k) - \beta\mathbb{E}\langle \nabla\Phi(x_k), \widehat{\nabla}\Phi(x_k) \rangle + L_\phi\beta^2\mathbb{E}\|\nabla\Phi(x_k)\|^2$$

$$+ L_\phi\beta^2\mathbb{E}\|\widehat{\nabla}\Phi(x_k) - \nabla\Phi(x_k)\|^2. \tag{66}$$

Also, based on eq. (41), we have

$$-\mathbb{E}\langle \nabla\Phi(x_k), \widehat{\nabla}\Phi(x_k) \rangle = -\mathbb{E}\langle \nabla\Phi(x_k), \widehat{\nabla}\Phi(x_k) - \nabla\Phi(x_k) \rangle - \mathbb{E}\|\nabla\Phi(x_k)\|^2$$

$$= \mathbb{E}\left[ -\langle \nabla\Phi(x_k), \mathbb{E}_k\widehat{\nabla}\Phi(x_k) - \nabla\Phi(x_k) \rangle \right] - \mathbb{E}\|\nabla\Phi(x_k)\|^2$$

$$\leq \mathbb{E}\|\nabla\Phi(x_k)\| \|\mathbb{E}_k\widehat{\nabla}\Phi(x_k) - \nabla\Phi(x_k)\| - \mathbb{E}\|\nabla\Phi(x_k)\|^2$$

$$\leq M_\Phi\mathbb{E}\|\mathbb{E}_k\widehat{\nabla}\Phi(x_k) - \nabla\Phi(x_k)\| - \mathbb{E}\|\nabla\Phi(x_k)\|^2,$$

which, in conjunction with eq. (66), yields

$$\mathbb{E}\Phi(x_{k+1}) \leq \mathbb{E}\Phi(x_k) + \beta M_\Phi\mathbb{E}\|\mathbb{E}_k\widehat{\nabla}\Phi(x_k) - \nabla\Phi(x_k)\| - \left(\beta - L_\phi\beta^2\right)\mathbb{E}\|\nabla\Phi(x_k)\|^2$$

$$+ L_\phi\beta^2\mathbb{E}\|\widehat{\nabla}\Phi(x_k) - \nabla\Phi(x_k)\|^2. \tag{67}$$

Using the bounds established in Lemma 9 (along with Jensen's inequality) and Proposition 8, we have

$$\mathbb{E}\Phi(x_{k+1}) \leq \mathbb{E}\Phi(x_k) + \beta M_\Phi\sqrt{\mathcal{B}_1} - \beta\left(1 - L_\phi\beta\right)\mathbb{E}\|\nabla\Phi(x_k)\|^2 + L_\phi\beta^2(\mathcal{B}_1 + \Delta).$$

Summing up the above inequality over $k$ from $k=0$ to $k=K-1$ yields

$$\mathbb{E}\Phi(x_K) \leq \mathbb{E}\Phi(x_0) + \beta K M_\Phi\sqrt{\mathcal{B}_1} - \beta\left(1 - L_\phi\beta\right)\sum_{k=0}^{K-1}\mathbb{E}\|\nabla\Phi(x_k)\|^2 + L_\phi K\beta^2(\mathcal{B}_1 + \Delta).$$

Setting $\beta = \frac{1}{L_\phi\sqrt{K}}$ and rearranging the above inequality yield

$$\frac{1 - \frac{1}{\sqrt{K}}}{K}\sum_{k=0}^{K-1}\mathbb{E}\|\nabla\Phi(x_k)\|^2 \leq \frac{(\Phi(x_0) - \Phi^*)L_\Phi}{\sqrt{K}} + M_\Phi\sqrt{\mathcal{B}_1} + \frac{\mathcal{B}_1 + \Delta}{\sqrt{K}}.$$

Setting $Q = \mathcal{O}(1)$, $D_f = \mathcal{O}(1)$, and $\mu = \mathcal{O}\left(\frac{1}{\sqrt{K dp^3}}\right)$ in the expressions of $\mathcal{B}_1$ and $\Delta$ completes the proof. $\qquad\square$