# OpenReview forum: "On the Convergence Theory for Hessian-Free Bilevel Algorithms"
_NeurIPS.cc/2022/Conference — NeurIPS 2022 Accept_

### Official Review · Reviewer_2qAw · 2022-07-09

**Rating:** 7
**Confidence:** 4
**Soundness:** 3 good
**Presentation:** 2 fair
**Contribution:** 3 good

**Summary:**

This work focuses on the Hessian-free bilevel optimization, which is a timely topic in machine learning, and has been widely used in many important applications such as neural architecture search, hyperparameter optimization, reinforcement learning, etc. In particular, the authors study the convergence performance and efficiency of Hessian-free bilevel algorithms via zeroth-order estimation, which have shown promise in many deep learning applications such as ES-MAML in meta-reinforcement learning, HOZOG in hyperparameter optimization. The authors first address an important variance and convergence issue of existing zeroth-order method, and propose to estimate only the response Jacobian of the hypergradient using direct forward propagation without any backpropagation. They further provide a convergence rate analysis for the resulting Hessian-free PZOBO method with partial hypergradient estimation and its stochastic version PZOBO-S. Extensive experiments support the convergence theories and justify the effectiveness of the designs in various applications.

**Questions:**

This method uses the gradient descent for both the inner and outer updates. It would be good to know if the accelerated gradient methods such as Nesterov’s momentum can be used to further improve the convergence and complexity performance.

The algorithm allows multiple random smoothing vectors, i.e., Q>1, which may be useful for reducing the estimation variance. I see that all experiments adopts the choice Q=1. Can the authors provide an explanation on why this choice is good? Furthermore, an experiment  on comparison of different Q should be provided.

Please add the units to x-axis of Figure 2(a), 2(b), and 2(c). For (a) and (b), the running time scales with 5, 10, 15, 20, but (c) scales with 0.5, 1, 1.5, 2. Please make the consistence accordingly.


**Ethics Review Area:**

["I don’t know"]

**Limitations:**

authors adequately addressed the limitations and potential negative societal impact of their work



**Strengths And Weaknesses:**

This work is generally written and motivated well. The finding that partial zeroth-order hypergradient estimation can improve the variance and convergence is a good contribution to Hessian-free bilevel optimization, in particular for the supervised scenarios. The proposed zeroth-order-like response Jacobian estimator enables a simple implementation in practice, in particular admitting a very easy extension to the stochastic case. This is kind of important because for conventional AID or ITD types of methods, more involved procedures such as Taylor expansion types of stochastic estimation are needed for hypergradient computations.

The analysis developed in this paper seems to be the first convergence rate analysis for zeroth-order bilevel optimization. Although the high-level idea comes from the zeroth-order optimization, there are indeed some differences under this bilevel structure. For example, from my reading, the smoothness property of the N-step output of the lower-level gradient-based algorithm established in Proposition 1 needs to conduct a recursive analysis on the entire gradient path, and the error characterization of stochastic Jacobian estimator needs to take the randomness of points in the path into consideration. Such analysis seems nontrivial to me.

Experiments seem to be comprehensive, on different applications such as representation learning, meta-learning, hyperparameter optimization. The proposed method indeed outperforms other bilevel optimizers quite a lot, in particular when the network size is large as in the meta-learning experiments.

---

> ### Author Response · Authors · 2022-08-02
> **Response to Reviewer 2qAw**
>
> Many thanks for providing the helpful review. In the revised version, we have made the changes based on the reviewer’s comments. All the changes are highlighted by the blue-colored texts.
>
> Q1: This method uses the gradient descent for both the inner and outer updates. It would be good to know if the accelerated gradient methods such as Nesterov’s momentum can be used to further improve the convergence and complexity performance.
>
> A: Great point! For the deterministic algorithm PZOBO, we anticipate that using acceleration methods can reduce the convergence error term $(1-\alpha\mu_g)^N$ in Theorem 1 to $(1-\sqrt{\alpha\mu_g})^N$,  and hence will reduce complexity. We will leave this interesting topic as our further study.
>
> Q2: The algorithm allows multiple random smoothing vectors, i.e., $Q>1$, which may be useful for reducing the estimation variance. I see that all experiments adopt the choice $Q=1$. Can the authors provide an explanation on why this choice is good? Furthermore, an experiment on comparison of different Q should be provided.
>
> A: Good point! We find that increasing the number of explorations $Q$ hurts efficiency (because more rounds of inner gradient updates are needed) and provides only marginal performance improvement in the experiments. We also find that the choice $Q=1$ achieves the best efficiency with nearly the same accuracy as other choices of $Q > 1$. We have included such a comparison in Fig. 7 in Appendix D of the revised pdf.
>
> Q3: Please add the units to x-axis of Figure 2(a), 2(b), and 2(c). For (a) and (b), the running time scales with 5, 10, 15, 20, but (c) scales with 0.5, 1, 1.5, 2. Please make the consistence accordingly.
>
> A: Thanks! We have made the x-axis scales of all plots to be consistent in the revised pdf.

---

> > ### Comment · Reviewer_2qAw · 2022-08-07
> > **response**
> >
> > Thanks for the clarification. My concerns have been fully addressed.

---

### Official Review · Reviewer_eYhq · 2022-07-11

**Rating:** 7
**Confidence:** 3
**Soundness:** 4 excellent
**Presentation:** 4 excellent
**Contribution:** 3 good

**Summary:**

The paper presents an algorithm and its stochastic extension to finite sums for solving smooth bilevel optimization problems which avoid computing the hessian associated to the lower level problem by instead using zeroth order oracles. Rather than approximate the entire gradient of the entire loss function in the upper level problem using zeroth order methods, the proposed algorithms only approximate the gradient of the solution mapping for the lower level problem with zeroth order methods, reducing variance and improving the method. Convergence guarantees are proven and four numerical experiments are performed that suggest the method is competitive with some contemporary methods.

**Questions:**

It's not clear what is the dimension of the hyperparameter vector in the various experiments; is p=1 in 4.1, for example? Even after searching the appendix I couldn't find it written what is the dimension of the hyperparameter vector for the applications.

On line 265 a bilevel optimization problem for an application is given - is the lower level problem really strongly convex and smooth here?

On line 131 you use that g is twice differentiable but you have only assumed it's C1?

**Limitations:**

One limitation is that there are many assumptions to apply the results. The assumptions are clearly stated which help alleviate this issue but the authors should address the limitations imposed by the assumptions in the numerical experiments because not made clear that the assumptions needed to apply the theorems are met. I think you should be explicit when/if the assumptions are being broken and whether or not the theorems that were proved actually apply to these practical examples. In the appendix it's sometimes mentioned when the lower level problem is strongly convex but I think for the neural network examples it could be nonsmooth and this should be mentioned as a limitation.

**Strengths And Weaknesses:**

Overall, I found the paper to be well written and of high quality. The main idea, to use zeroth approximations of the Jacobian of the solution mapping for the lower level problem with respect to the hyperparameters appears to be original and significant, to the best of my knowledge. Some parts of the paper could be clearer, as touched on in the questions and limitations sections, but otherwise the main ideas are well-communicated. Regarding the experiments, they are limited to comparing to baselines and I think it's a weakness not to compare against other first-order optimization methods like AmIGO [Arbel, Mairal 2022] or single loops algorithms like SABA [Dagréou et al 2022].

"Amortized Implicit Differentiation for Stochastic Bilevel Optimization" - Michael Arbel, Julien Mairal, 2022

"A framework for bilevel optimization that enables stochastic and global variance reduction algorithms" - Mathieu Dagréou, Pierre Ablin, Samuel Vaiter, Thomas Moreau, 2022

---

> ### Author Response · Authors · 2022-08-02
> **Response to Reviewer eYhq**
>
> Many thanks for providing the helpful review. In the revised version, we have made the changes based on the reviewer’s comments. All the changes are highlighted by the blue-colored texts.
>
> Q1: It's not clear what is the dimension of the hyperparameter vector in the various experiments; is $p=1$ in 4.1, for example? Even after searching the appendix I couldn't find it written what is the dimension of the hyperparameter vector for the applications
>
> A: Sorry for the confusion! For the hyper-representation problems, the dimension of the hyperparameter vector (outer variable) corresponds to the number of parameters in the embedding model. For example, for the deep hyper-representation problem, the dimension of the hyperparameter vector $p$ corresponds to the number of weights in the backbone LeNet network. For the shallow hyper-representation problem, the value of $p$ depends on the input dimension and representations dimension d. For this experiments we always set the input dimension to be half of the representations dimension d, which corresponds to $p = d \times d / 2$ for the linear embedding settings.
>
> For meta-learning experiments, the dimension of the hyperparameter vector (or outer variable) corresponds to the number of weights in the common embedding model (which is either ResNet12 or CNN4). For hyperparameter optimization experiments, we used one regularization parameter for each weight in the linear classification model, which means that we have as many hyperparameters as parameters. The dimension of the input features in the 20 Newsgroup dataset is 130K and we simply use a linear model without bias term, which corresponds to the number of parameters (and thus hyperparameters) $p=130K \times 20$.
>
> We have added the above specification in the revised pdf.
>
> Q2: On line 265 a bilevel optimization problem for an application is given - is the lower level problem really strongly convex and smooth here?
>
> A: Yes. It can be seen that this lower-level problem contains a loss term that is quadratic w.r.t. the variable $w$ (hence convex and smooth w.r.t. $w$) plus a strongly-convex square regularization $\frac{\gamma}{2}\|w\|^2$. Therefore, the overall lower-level problem is strongly-convex and smooth.
>
> Q3: On line 131 you use that $g$ is twice differentiable but you have only assumed it's C1?
>
> A: Sorry for the confusion! In the revised pdf, we have clarified that the $g$ function is twice differentiable right after eq. (1).
>
> Q4: The assumptions are clearly stated which help alleviate this issue but the authors should address the limitations imposed by the assumptions in the numerical experiments because not made clear that the assumptions needed to apply the theorems are met. I think you should be explicit when/if the assumptions are being broken and whether or not the theorems that were proved actually apply to these practical examples. In the appendix it's sometimes mentioned when the lower level problem is strongly convex but I think for the neural network examples it could be nonsmooth and this should be mentioned as a limitation.
>
> A: Good point! For the experiments in Sections 4.1, 4.4, the bilevel problems are relatively simpler with quadratic or logistic loss with a linear classifier. It can be checked that the strong-convexity, smoothness properties are satisfied. For the experiments that involve neural networks, e.g., in deep hyper-representation (Section 4.2) and in meta-learning (Section 4.3), the lower-level problem optimizes only the last-layer parameters (hence is linear) and hence is still strongly-convex and smooth. However, the smoothness of the upper-level problem is not guaranteed, e.g., for ReLU activations, which may not be a big issue since we can use some smoothed ReLU variants. We have clarified this discussion in Append B in the revised pdf.

---

> > ### Comment · Reviewer_eYhq · 2022-08-07
> > **Clarifications sufficient**
> >
> > All of my concerns are addressed by the rebuttal and the revisions and I appreciate the authors' thorough response. I have updated the presentation score to a 4.

---

### Official Review · Reviewer_mkCb · 2022-07-12

**Rating:** 5
**Confidence:** 5
**Soundness:** 4 excellent
**Presentation:** 2 fair
**Contribution:** 2 fair

**Summary:**

## Further Update

Not all my concerns are fairly treated and addressed in the discussion period (as the authors have promised). I lower the overall rating from 6 to 5 and the presentation rating from 3 to 2. This is my final decision.

## Update

I changed my rating from 4 to 6 due to the fair rebuttal from the authors.

---

This paper proposes a zeroth-order based solver for bi-level optimization problem. In order to avoid the heavy computation of Hessian matrix and its inversion, the paper proposes to directly approximate the Jacobian matrix (implicit gradient) through function value difference and develops a Hessian-free bi-level algorithm. For the proposed method, the authors provide convergence analysis for the proposed zeroth-order algorithm and reach a non-asymptotic performance guarantee.

**Questions:**

Please see the comments in the weaknesses above.

**Limitations:**

The authors have adequately addressed the limitations and potential negative societal impact.

**Strengths And Weaknesses:**

Strengths:

1. Theoretical: The theoretical contributions are the main strength of this paper. To analyze the zeroth-order approximation, the authors must take into consideration the bias, variance, smoothness, and boundedness of the proposed optimizer, and this procedure is even more difficult when the commonly adopted Lipschitz-smooth assumption does not apply. The authors make very detailed discussion on different application scenarios of the proposed method and obtain meaningful result. This is difficult and remarkable.

2.  Various experiments were conducted on different deep learning tasks to show the effectiveness of the proposed methods. The results generally look good and do outperform baselines.

Weaknesses:

1. Novelty: My main concern lies in the novelty of the paper. As is all known, that zeroth-order is a common technique even in the context of bi-level optimization. For example, DARTS [34] also used the zeroth-order method to solve the Jacobian-involved term without calculating Hessian matrix. I wonder what is the difference between this paper and DARTS in terms of theory as well as performance. In general, zeroth-order approach is not a novel ingredients in bi-level optimization and this greatly constrains the novelty of the proposed method.

2. Experiments: I strongly suggest the authors to add some experiments on neural architecture search (NAS) for bi-level tasks. As I mentioned above, DARTS [34] used very similar approaches and I am curious which one performs better.

3. Multiple trials: The authors do not report any results on the variance of any of the experiments. It is significance to report the variance to show the stability of each algorithm as well as the significance of the improvement.

4. Minor issues: In all the plots/figures, the authors are suggested to use similar font sizes in ticks, axis labels, and legends. It is difficult to read the labels of all the figures in this paper.

In general, I believe the theoretical part of the paper is excellent, but the method proposed lacks major innovation and significance.

---

> ### Author Response · Authors · 2022-08-02
> **Response to reviewer mkCb**
>
> Many thanks for providing the helpful review. In the revised version, we have made the changes based on the reviewer’s comments. All the changes are highlighted by the blue-colored texts.
>
> Q1. Novelty: My main concern lies in the novelty of the paper. As is all known, that zeroth-order is a common technique even in the context of bi-level optimization. For example, DARTS [34] also used the zeroth-order method to solve the Jacobian-involved term without calculating Hessian matrix. I wonder what is the difference between this paper and DARTS in terms of theory as well as performance. In general, zeroth-order approach is not a novel ingredients in bi-level optimization and this greatly constrains the novelty of the proposed method.
>
> A: Although zeroth-order optimization has been used in bilevel optimization, both the design of our zeroth-order-like response Jacobian in eq. (4) and its use to partially estimating the hypergradient are new to the literature. Let us elaborate on this below.
>
> 1. Comparison to DARTS [34]: our proposed method has substantial differences from the zeroth-order-like estimation in DARTS [34] (i.e., eq. (8) therein).  We have the following differences in the related work of the revised pdf.
>
>     * In terms of the estimator design, DARTS estimates the matrix-vector product (using our notations) $\nabla_x\nabla_yg(x,y)\nabla_yf(x,y^1)$ with $y^1=y-\alpha\nabla g(x,y)$. In contrast, our problem estimates the $\textbf{response Jacobian matrix}$ $\nabla_x\nabla_yg(x,y^N)(\nabla_y^2g(x,y^N))^{-1}$. Here, the Hessian matrix $\textbf{inverse}$ is more challenging to estimate than the matrix $\textbf{without inverse}$ in DARTS.
>
>     * Second, the estimator in DARTS uses the $\textbf{value}$ difference $\frac{\nabla_x g(y+\epsilon\nabla_yf(x,y^1))-\nabla_x g(y-\epsilon\nabla_yf(x,y^1))}{2\epsilon}$ at two points with a gap of the gradient $\epsilon\nabla_yf(x,y^1)$. In contrast, we use the $\textbf{iterate}$ difference $\frac{y^N(x+\epsilon u)-y^N(x)}{\epsilon}u^T$ at two points with gap of a Gaussian random vector $\epsilon u$. Also, our estimator is multiplied with a Gaussian random vector $u$, which does not exist in DARTS.
>
>     * Third, our estimator applies to multiple inner steps with $N>1$ with performance guarantee, whereas the estimator eq. (8) used in DARTS applies to only the one-step case with $N=1$ and its extension to multi-step case seems nontrivial because the gradient of eq. (6) with multiple inner steps will induce products of second-order matrices.
>
>     * In theory, since DARTS uses a one-step gradient descent to estimate the inner minimizer $y^*(x)$, which can introduce nonvanishing error and divergence in some scenarios. In addition, their zeroth-order-like estimator has no performance guarantee so far for the bilevel objective. As a comparison, our method is guaranteed to converge due to the more flexible choice of $N$ as well as a provable property of the smoothness.
>
> 2. As also mentioned in the introduction, our partially zeroth-order-like estimation is different from existing zeroth-order bilevel methods such as ES-MAML, HOZOG, etc.
>
> Q2: Experiments: I strongly suggest the authors to add some experiments on neural architecture search (NAS) for bi-level tasks. As I mentioned above, DARTS [34] used very similar approaches and I am curious which one performs better.
>
> A: Thanks for your great suggestions! We have added a new experiment (see Fig. 6 in Appendix C) to compare with DARTS in the revised pdf. It can be seen that our method PZOBO-S finds a better descent direction, decreases the objective more aggressively at each step, and attains a lower loss value. More results will be added.
>
> Q3: Multiple trials: The authors do not report any results on the variance of any of the experiments. It is significance to report the variance to show the stability of each algorithm as well as the significance of the improvement.
>
> A: Thanks! We have plotted the covariances for the experiments in Fig. 2, Fig. 3 and Fig. 5.  The covariances in Fig. 3 (a) and 3(b) are easy to observe. For Fig. 2 and Fig. 5, since the experiments are relatively at small scale with a linear classifier, the performance variation of each algorithm is small and their covariances are hard to observe. We have commented on this for clarification in the revised pdf.
>
> For the meta-learning experiments with larger datasets (miniImageNet) and networks (ResNet), one run can take a very long time, e.g., MetaOptNet takes several days to achieve 70% accuracy, and we may not be able to finish the covariance plot within this response period. However, we will add the covariance results in the figure once we finish.
>
> Q4: Minor issues: In all the plots/figures, the authors are suggested to use similar font sizes in ticks, axis labels, and legends. It is difficult to read the labels of all the figures in this paper.
>
> A: Good point! We have followed your suggestion to improve our plots in the revised pdf.

---

> > ### Author Response · Authors · 2022-08-02
> > **Response to reviewer mkCb (continue)**
> >
> > Finally, we thank the reviewer again for the helpful comments for our work. If our response resolves your concerns to a satisfactory level, we kindly ask the reviewer to consider raising the rating of our work. Certainly, we are more than happy to address any further questions you may have during the discussion period.

---

> > ### Comment · Reviewer_mkCb · 2022-08-04
> > **Comments on "Response to reviewer mkCb"**
> >
> > Thanks very much for the explanation and additional experiments. I have the following comments.
> >
> > 1. Thanks for the A1 to distinguish the difference between your work and DARTs-related work. I strongly suggest to highlight this part in your revision (suppose more pages are allowed), since for people who know and work on BLO, your work could be underestimated.
> > 2. In Q4 I am sorry not to make myself clear. I mean by standard, all the texts in the figures (legends, labels, axis-ticks, annotations) should be of the similar font size to that of the texts in the paper. The current font is still too small and does not match the font size of the texts in the paper. This is for the readability as well as visual effects.
> >
> > Finally, I think the author has adequately addressed my concerns and I agree to change my rating to 6. The reason I does not go with 7 like other reviewers is I still think the novelty in A1 does not touch the deep essence. However, this does not cast a big shadow on this work.
> >
> > Thanks.

---

> > > ### Author Response · Authors · 2022-08-05
> > > **Thanks for the further comments!**
> > >
> > > We thank the reviewer very much for the further feedback and for increasing the rating. We will move the discussion about the difference between our work and the work on DARTS to the main body of the paper in our final version. Also many thanks for further clarifying your suggestion on the format of the figures. We will make the changes accordingly.

---

### Official Review · Reviewer_vjNx · 2022-07-12

**Rating:** 7
**Confidence:** 3
**Soundness:** 4 excellent
**Presentation:** 3 good
**Contribution:** 3 good

**Summary:**

This paper introduces a Hessian-free bi-level optimizer called partial zeroth-order-like bilevel optimizer (PZOBO). The method uses a zeroth-order-like Jacobian estimator, which provides accurate hypergradient estimates and a computationally effective way of solving bi-level optimization problems. The paper presents a thorough theoretical analysis and experimental validation on various bi-level problems.

**Questions:**

+ Covariances should be added to the curves to better illustrate the performance variation. For example, Figures 3(a) and 3(b) shows the covariances but other figures do not.

+ In related work, an analysis of the limitations of previous solvers and a discuss of how the proposed method can address those limitations can be helpful to evaluate the novelty and impact of the proposed method.

+ The text refers to Figure 4(a), (b) and (c), but Figure 4 does not have subfigure labels.

+ The consistency of figure formatting can be improved. For example, the shadows at the bottom of some figures such as Figures 2(d) and Figures 2(f).


**Limitations:**

This work does not have negative society impacts. A further discussion of the limitations of the proposed bi-level optimizer will be helpful. For example, the paper assumes the inner-level function to be strongly convex. How the proposed approach can be extended to address problems without this assumption?

**Strengths And Weaknesses:**

+ The paper addresses the important bi-level optimization problem by providing an efficient and accurate solution.

+ The paper shows the convergence guarantee of the proposed approach.

+ Experimental results and comparisons with various baseline bi-level optimizers are provided over various bi-level problems. The proposed method generally outperforms baseline methods, especially in high-dimensional problems.

---

> ### Author Response · Authors · 2022-08-02
> **Response to Reviewer vjNx**
>
> Many thanks for providing the helpful review. In the revised version, we have made the changes based on the reviewer’s comments. All the changes are highlighted by the blue-colored texts.
>
> Q1: Covariances should be added to the curves to better illustrate the performance variation. For example, Figures 3(a) and 3(b) show the covariances but other figures do not.
>
> A: Thanks! We have plotted the covariances for Fig. 2 and Fig. 5, but since their experiments are relatively small scale with a linear classifier, the performance variation of each algorithm is very small and their covariances are hard to observe. We have commented on this for clarification in the revised pdf.
>
> For the meta-learning experiments with larger datasets (miniImageNet) and networks (ResNet), one run can take a very long time, e.g., MetaOptNet takes several days to achieve 70% accuracy, and we may not finish the covariance plot within this response period. However, we will add the covariance results in the figure once we finish.
>
> Q2: In related work, an analysis of the limitations of previous solvers and a discussion of how the proposed method can address those limitations can be helpful to evaluate the novelty and impact of the proposed method.
>
> A: Great suggestion! We have added this discussion in the related work of the revised pdf.
>
> Q3: The text refers to Figure 4(a), (b) and (c), but Figure 4 does not have subfigure labels.
>
> A: Thanks! We have added subfigure labels in Fig. 4 in the revised pdf.
>
> Q4: The consistency of figure formatting can be improved. For example, the shadows at the bottom of some figures such as Figures 2(d) and Figures 2(f).
>
> A: Thanks! We have unified all figure formats in the revised pdf.

---

### Author Response · Authors · 2022-08-02
**Response to all reviewers**

Many thanks for providing the helpful reviews. In the revised version, we have made the changes based on the reviewers’ comments. All the changes are highlighted by the blue-colored texts. Our major changes in the revised pdf  are summarized as below.

1. We have added the following experimental results:

    * We have added a plot (Fig. 7 in Appendix D) on the comparison of our methods with different choices of $Q$.

    *  We have added a new experiment (Fig. 6 in Appendix C) to compare our PZOBO-S  method with DARTS.

    *  We have unified the formats (size, units, axis scale, legend, etc) for all experiments.

    *  We have added a discussion (Appendix B) of the consistency between our assumptions and our experimental problems.

    *  We have specified the problem dimensions (Appendix G) for all experimental setups.

2. In Section 1.1 of Related Work, we have added a detailed discussion on the comparison of our method to existing bilevel algorithms and how our novel designs can address the drawbacks of these methods.

3. In Appendix C.1, we have provided a detailed comparison between our method and DARTS [35], and explained that our proposed zeroth-order-like Jocabian estimator has substantial differences from the estimator in DARTS [35] in terms of design and theory.

---

### Meta-Review · Area_Chair_YGQf · 2022-08-23

**Recommendation:** Accept
**Confidence:** Certain

**Metareview:**

The paper introduces a Hessian-free bi-level optimizer called partial zeroth-order-like bilevel optimizer (PZOBO). PZOBO uses a zeroth-order-like Jacobian estimator, which provides accurate hypergradient estimates and a computationally effective way of solving bi-level optimization problems. The paper presents a thorough theoretical analysis and experimental validation on various bi-level problems.

Strengths:

1 - The authors address a relevant problem and provide an efficient and accurate solution.
2 - Convergence guarantees are provided for the proposed approach.
3 - Experimental results and comparisons with baselines over various bi-level problems.
4 - PZOBO generally outperforms baseline methods, especially in high-dimensional problems.
5 - The main idea, to use zeroth approximations of the Jacobian of the solution mapping for the lower level problem with respect to the hyperparameters appears to be original and significant.

Weaknesses:

1 - One limitation is that there are many assumptions to apply. The assumptions are clearly stated which help alleviate this issue but the authors should address the limitations imposed by the assumptions in the numerical experiments because not made clear that the assumptions needed to apply the theorems are met.

Decision:

Overall, all the reviewers vote for acceptance. This is a strong paper with very few limitations and with a significant contribution to the field. Because of this, I recommend acceptance. I encourage the authors to follow the reviewers' comments in order to improve the paper for the camera-ready version.



**Award:**

No

---

### Decision · Program_Chairs · 2022-09-14

Accept